# Real-Time Aligned Reward Model beyond Semantics

Zixuan Huang [1 2]  Xin Xia [2]  Yuxi Ren [2]  Jianbin Zheng [2]  Xuefeng Xiao [2]  Hongyan Xie [1]  Li Huaqiu [3]
Songshi Liang [4]  Zhongxiang Dai [5]  Fuzhen Zhuang [6]  Jianxin Li [1]  Yikun Ban [1 *]  Deqing Wang [1 †]

## Abstract

Reinforcement Learning from Human Feedback (RLHF) is a pivotal technique for aligning large language models (LLMs) with human preferences, yet it is susceptible to reward overoptimization, in which policy models overfit to the reward model, exploit spurious reward patterns instead of faithfully capturing human intent. Prior mitigations primarily rely on surface semantic information and fails to efficiently address the misalignment between the reward model (RM) and the policy model caused by continuous policy distribution shifts. This inevitably leads to an increasing reward discrepancy, exacerbating reward overoptimization. To address these limitations, we introduce **R2M** (**R**eal-Time Aligned **R**eward **M**odel), a novel lightweight RLHF framework. R2M goes beyond vanilla reward models that solely depend on the semantic representations of a pretrained LLM. Instead, it leverages the evolving hidden states of the policy (namely **policy feedback**) to align with the real-time distribution shift of the policy during the RL process. This work points to a promising new direction for improving the performance of reward models through real-time utilization of feedback from policy models.

## 1. Introduction

Reinforcement Learning from Human Feedback (RLHF) has become a cornerstone technique for aligning large language models (LLMs) with human values and preferences (Vemprala et al., 2023; Huang et al., 2025; Shen & Zhang, 2024; Shen et al., 2025; Hu et al., 2024). However, RLHF faces a persistent challenge: reward overoptimization. In-stead of faithfully capturing human intent, policy models often exploit spurious reward patterns, such as response length, markdown formatting, or superficial linguistic cues like certain n-grams or emojis, to maximize rewards without genuinely improving alignment (Gao et al., 2023; Coste et al., 2023; Eisenstein et al., 2023). The core issue lies in the reward model: trained on limited preference data, it can only approximate human values. As the policy evolves during RLHF training while the reward model remains fixed, distribution shift exacerbates approximation errors (Wang et al., 2024b), ultimately leading to unreliable reward signals in optimization.

A common mitigation is to iteratively update the reward model so that it adapts to the policy's evolving behavior. Yet, direct retraining of the reward model at each iteration is computationally prohibitive. To address this, one research direction emphasizes uncertainty-aware corrections. Coste et al. (2023); Eisenstein et al. (2023); Zhai et al. (2023) penalize uncertain samples during policy training, while Zhang et al. (2024a) introduce kernel-based uncertainty estimates derived from reward model embeddings. Another line of work focuses on robust reward model retraining. Lang et al. (2024) incorporate an unsupervised mutual information loss to counter distribution shift, and Liu et al. (2024) augment training data by decomposing preferences relative to prompts. These methods trade off efficiency and robustness, but leave open a critical question: *Can we design a new RLHF framework that preserves training efficiency while effectively achieving real-time alignment of reward models towards policy models?*

We propose **R2M** (**R**eal-Time Aligned **R**eward **M**odel) to solve this challenge. It is a lightweight RLHF framework in which the RM itself is reinforced iteratively by dynamically adapting to the policy's internal states, and it does not require any additional labeled data or environmental feedback to improve the performance.

Specifically, we observe that deep-layer hidden states of the policy encode latent patterns that are closely correlated with both golden human preferences and the scalar reward scores assigned by RMs. This observation aligns with the perspectives on implicit reward modeling advanced in works such as DPO (Rafailov et al., 2023) and PRIME (Cui et al.,

---
[*]Co-advised this work.  [1]Beihang University [2]Bytedance China [3]Tsinghua University [4]Renmin University of China [5]The Chinese University of Hong Kong, Shenzhen [6]School of Artificial Intelligence, Beihang University. Correspondence to: Deqing Wang <dqwang@buaa.edu.cn>.

*Proceedings of the 43rd International Conference on Machine Learning*, Seoul, South Korea. PMLR 306, 2026. Copyright 2026 by the author(s).

2025), yet it is often overlooked by existing explicit reward models (Liu et al., 2024; Zhang et al., 2024a).

Building on this insight, we aim to go beyond reward models that solely depend on semantic representations of a pre-trained LLM. Instead, we enhance the reward model by incorporating the evolving hidden states of the policy model (namely **policy feedback**). To this end, we redesign the scoring head of the RM so that it dynamically integrates these hidden states, enabling the RM to adapt to distribution shifts in the policy. In our RLHF framework, this introduce a lightweight training component that learns to aggregate policy feedback directly, enhancing the RM's representation without retraining the entire model. Owing to its efficiency, this mechanism can be seamlessly applied at every training round, ensuring continuous synchronization between the reward model and the policy model.

The design of R2M offers two benefits: **1) Iterative distribution alignment with accurate reward allocation.** The reward model integrates the policy's evolving hidden states which provide behaviorally grounded and semantically informed feedback. This mitigates distribution shifts, reduces reward overoptimization, and ensures more accurate reward assignment. **2) Extremely lightweight overhead.** R2M only need to learn how to aggregate representations, introducing negligible additional cost.

Experimental results demonstrate that R2M significantly improves performance on dialogue tasks (trained on UltraFeedback (Cui et al., 2023), evaluated on Alpaca-Eval (Dubois et al., 2024)) and text summarization tasks (trained and evaluated on TL;DR summarization dataset). Specifically, compared with vanilla RLOO, RLOO+R2M increases the AlpacaEval 2 win rate (WR) by 5.2% - 8.0%, the length-controlled win rate (LC) by 2.9% - 6.1% and the TL;DR win rate by 6.3% compared to baselines, while introducing only minimal computational cost. Furthermore, we conducted a comprehensive analysis, showing that R2M effectively strengthens the vanilla RM and mitigates reward overoptimization with minimal additional training overhead.

## 2. Preliminary

RLHF consists of three main steps: **1)** Supervised Fine Tuning, **2)** Reward Modeling, and **3)** RL optimization. We provide a detailed workflow shown in Appendix H.1. As R2M is designed to directly integrated into the RL optimization phase, let us consider the following typical third-stage RL Optimization process:

**Trajectory Sampling**: At each training step $t \in [T]$, we update offline policy $\pi_{old}$ to online policy $\pi_\theta$. Then, given a query set $X_t = \{x_1, x_2, \ldots, x_n\}$, $\pi_{old}$ is used to sample a group of $K$ responses $G_i = \{y_{i,j}\}_{j=1}^K$ for each $x_i \in X_t$.

**Reward Annotation**: For each $(x_i, G_i), i \in [n]$, there are $K$ query-response pairs $(x_i, y_{i,j}), j \in [K]$. We use a scalar reward model $r_\varphi(x, y)$ to assign scores to each query-response pair, obtaining $\{r_{i,j} | i \in [n], j \in [K]\}$, resulting in a batch $\mathcal{B} = \{(x_i, y_{i,j}, r_{i,j}) | i \in [n], j \in [K]\}$. After this process, we employ the RLOO approach (Ahmadian et al., 2024) to perform advantage estimation within each $G_i$:

$$\hat{A}_{i,j} = r_{i,j} - \frac{1}{K-1} \sum_{\hat{j} \neq j} r_{i,\hat{j}}. \tag{1}$$

**Policy Optimization**: For each query-response pair $(x_i, y_{i,j})$, we perform a forward pass in the policy model $\pi_\theta$ and optimize $\pi_\theta$ using importance sampling by maximizing the following objective (Shao et al., 2024; Ahmadian et al., 2024), where $\varepsilon$ and $\beta$ are hyperparameters:

$$\begin{aligned} \min[&\frac{\pi_\theta(y_{i,j}|x_i)}{\pi_{\theta_{old}}(y_{i,j}|x_i)} \hat{A}_{i,j}, \\ &\text{clip}(\frac{\pi_\theta(y_{i,j}|x_i)}{\pi_{\theta_{old}}(y_{i,j}|x_i)}, 1 - \varepsilon, 1 + \varepsilon) \hat{A}_{i,j}] \\ &- \beta \mathbb{D}_{KL}[\pi_\theta \| \pi_{\text{ref}}] \end{aligned} \tag{2}$$

The design of R2M is based on the aforementioned RL optimization process. As a lightweight and significantly effective alternative, R2M can be seamlessly deployed to all REINFORCE-based RLHF frameworks. Due to resource constraints, we adopt RLOO as one of the primary baselines.

## 3. Motivation

We argue that deep-layer hidden states of the policy in a transformer's forward pass contain crucial information that are closely correlated with both golden human preferences and reward scores, making them effective for enhancing vanilla RMs. Due to space constraints, the experimental details of this section are provided in the Appendix G.1.

Figure 1 establishes the relationship between hidden state similarity and preference labels. The average hidden state similarity between pairs with different preference labels is significantly lower than that between pairs with the same preference label, and this gap widens progressively with increasing layer depth. This indicates that **deep-layer hidden states effectively capture human preferences**. Similar viewpoints have also been expressed in works on implicit RMs, such as DPO (Rafailov et al., 2023) and PRIME (Cui et al., 2025), yet this information beyond semantics is often overlooked by existing explicit RMs.

Figure 2 establishes the relationship between *deep-layer* hidden state similarity and the absolute difference of reward scores, they exhibit a strong negative correlation: **higher hidden state similarity corresponds to smaller reward**

**differences,** which is consistent with the observation in DPO (Rafailov et al., 2023): during the preference optimization process, the language model implicitly assumes the role of the reward model. This significant correlation further suggests the potential for effective alignment between the hidden states of the policy model and the reward model.

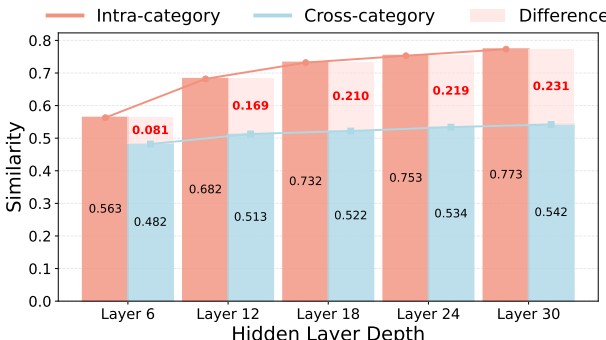

*Figure 1.* The average hidden state similarity of the same-preference pair set and the different-preference pair set across transformer layers. Each pair consists of two query-response samples with respective preference labels.

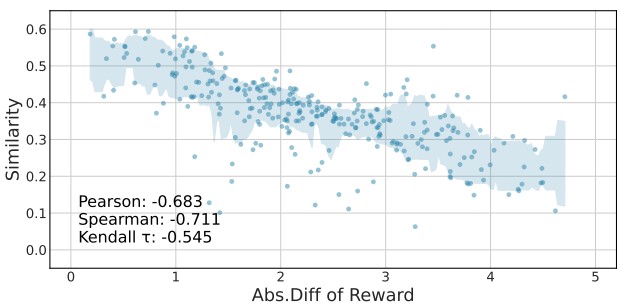

*Figure 2.* Negative correlation between absolute difference of reward scores allocated by the RM and hidden state similarity. Each data point corresponds to a query-response pair labeled with either identical or differing human preferences.

These findings strongly confirm that **a policy's hidden states offer valuable insights for alignment of RMs towards policy models.** Theorem 3.1 further shows that, when $\gamma^{(t)} > 0$, since $\left(1 - \gamma^{(t)}\right)^{1/2} < 1$, R2M yields a tighter upper bound of $\epsilon$ compared with the vanilla RM.

---

**Theorem 3.1.** *(Proof in Appendix A.1) Suppose that $\epsilon$ quantifies the extent of reward misalignment, we have the following upper bound of $\epsilon$ for R2M and vanilla RM:*

$$\epsilon_{R2M}^{(t)} \leq \left(1 - \gamma^{(t)}\right)^{1/2} \cdot C + \Delta \mathcal{D}^{(t)} \cdot L$$

$$\epsilon_{vanilla}^{(t)} \leq C + \Delta \mathcal{D}^{(t)} \cdot L$$

*where $\gamma^{(t)} \in [0, 1]$, and $C > 0$.*

---

## 4. Method

Figure 3 illustrates the overall workflow of R2M. Built upon vanilla RL optimization frameworks, R2M primarily addresses the following challenges: **1)** how to structurally incorporate feedback messages from the policy model into the reward model (Section 4.1); **2)** how to design the optimization objectives for the reward model (Section 4.2).

### 4.1. Reward Model Structure

In this section, we focus on integrating the policy feedback into the reward model. As shown in Figure 4, we introduce a policy feedback data flow that bypasses the LLM part to directly enhance the original Reward Token Embedding (introduced in Appendix H.1). We formally redefine the reward model $r_\varphi(x, y)$ with policy feedback $h$ as $r_\varphi(x, y, h)$. To effectively utilize the policy feedback, R2M contains two pivotal extra components: **Sequence-to-Token Cross Attention** and **Time-Step-Based Weighted Combination**.

Specifically, during *Trajectory Sampling*, we collect the last-layer hidden states $h_{i,j} \in \mathbb{R}^{S_{i,j}-1 \times D_p}$ for each query-response pair $(x_i, y_{i,j}), i \in [n], j \in [K]$ from the policy. Here, $S_{i,j}$ denotes the length of the query-response pair, and $D_p$ represents the hidden size of the policy. Then, during *Reward Annotation*, each $(x_i, y_{i,j})$ is fed into the reward model's LLM component to derive the Reward Token Embedding (RTE) $H_{\text{last}}^{i,j} \in \mathbb{R}^{1 \times D_{\text{rm}}}$ (denoted in Appendix H.1).

**Sequence-to-Token Cross Attention.** We introduce a cross-attention component to *extract relevant information from hidden states* of query-response pairs, while *bridging the semantic gap* between heterogeneous policy models and reward models (discussed in Appendix A.1). Specifically, we inject policy feedback by performing a cross-attention operation from the sequence to a single token. This enables the query of the RTE $q = H_{\text{last}}^{i,j} W_q$ to fully absorb the keys $k = h_{i,j} W_k$ and values $k = h_{i,j} W_v$ of the hidden state sequence $h_{i,j}$, which contains both policy state information and sequence semantic information, and updates it into a more information-rich Aggregated RTE:

$$\widehat{H}_{\text{last}}^{i,j} = \text{Softmax}\left(\frac{qk^T}{\sqrt{d}}\right) vW_o \in \mathbb{R}^{1 \times D_{\text{rm}}}, \quad (3)$$

where $W_q \in \mathbb{R}^{D_{\text{rm}} \times d}$, $W_k, W_v \in \mathbb{R}^{D_p \times d}$, and $W_o \in \mathbb{R}^{d \times D_{\text{rm}}}$ are learnable weight matrices of the cross-attention module, with $d$ representing the internal width.

**Time-Step-Based Weighted Combination.** After obtaining $\widehat{H}_{\text{last}}^{i,j}$, we adopt an exploration-exploitation approach (Ban et al., 2021; 2024; Huang et al., 2025) to balance the weights of $H_{\text{last}}^{i,j}$ and $\widehat{H}_{\text{last}}^{i,j}$, yielding the final RTE $H_{\text{fin}}^{i,j}$. Specifically, we use a time-step-based approach to gradually decrease the

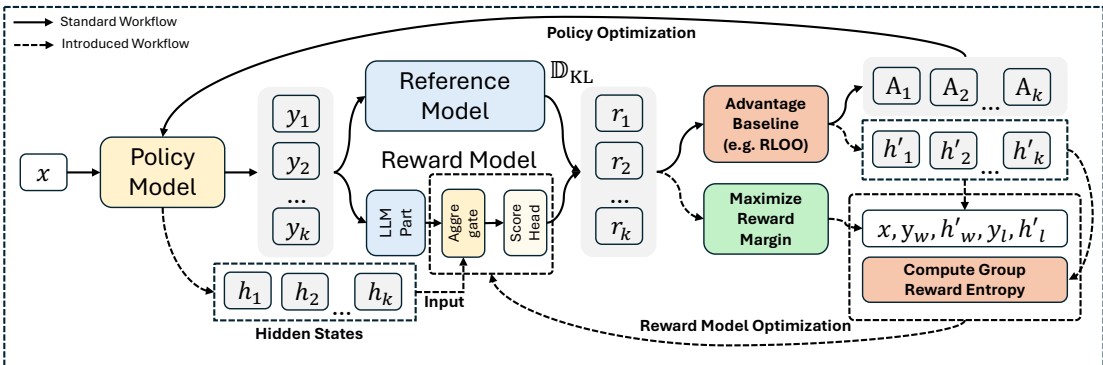

*Figure 3.* Overview of R2M. We first aggregate the last-layer hidden states $h_i$ from the policy with the LLM part output of the reward model. This aggregated representation is then fed into the scoring head for reward prediction. When the policy updates, we get the real-time feedback $h'_i$ and utilize it to construct preference pairs. Finally, we optimize the reward model by jointly minimizing the Bradley-Terry loss and the Group Reward Entropy loss.

weight on the original RTE $H_{\text{last}}^{i,j}$ as follows:

$$H_{\text{fin}}^{i,j} = (1 - \omega(t))\widehat{H}_{\text{last}}^{i,j} + \omega(t)H_{\text{last}}^{i,j},$$
$$\omega(t) = \max(\frac{1}{2}\cos(\frac{t}{T}\pi) + \frac{1}{2}, \Omega), \quad (4)$$

where $t$ is the current training round, $T$ is the total number of training rounds, $\Omega$ is the minimum weight of $H_{\text{last}}^{i,j}$, and $\omega(t)$ is a monotonically decreasing function of $t$ (Wu et al., 2025). When $t$ is small, we prioritize leveraging the existing RTE $H_{\text{last}}^{i,j}$. As R2M iteratively updates during the training process (as discussed in Section 4.2), we gradually increase the influence of $\widehat{H}_{\text{last}}^{i,j}$ to enable R2M to progressively identify and adapt to the distribution shift of the policy. As a result of balancing the exploitation of the original embedding with the exploration of policy feedback information, $H_{\text{fin}}^{i,j}$ is then mapped by the reward head $\phi$ to the final scalar reward $r_\varphi(x_i, y_{i,j}, h_{i,j}) = \phi(H_{\text{fin}}^{i,j}) \in \mathbb{R}$.

## 4.2. Iterative Reward Model Lightweight Optimization

In Section 4.1, we have introduced policy feedback into the RM. However, the semantic spaces are not yet aligned, making it challenging for the reward model to directly utilize this information. To address this, we incorporate an extra lightweight Reward Model Optimization phase following the Policy Optimization phase at each training step, and propose a novel optimization objective for R2M, namely the Group Reward Entropy Bradley-Terry (GREBT) loss.

**Hidden State Update**. To ensure that the hidden states $h_{i,j}$ remain up-to-date and accurately reflect the internal states of the policy $\pi_\theta$, we update $h_{i,j}$ whenever $(x_i, y_{i,j})$ is used to update $\pi_\theta$. Specifically, during the forward pass of $\pi_\theta$ on $(x_i, y_{i,j})$, we fetch the latest hidden states $h_{i,j}$, which incurs no additional computational overhead. Since the policy model is trained for $k$ epochs on the same batch at each training step $t$ (Shao et al., 2024; Hu, 2025), this update is performed only in the final epoch. For notational simplicity,

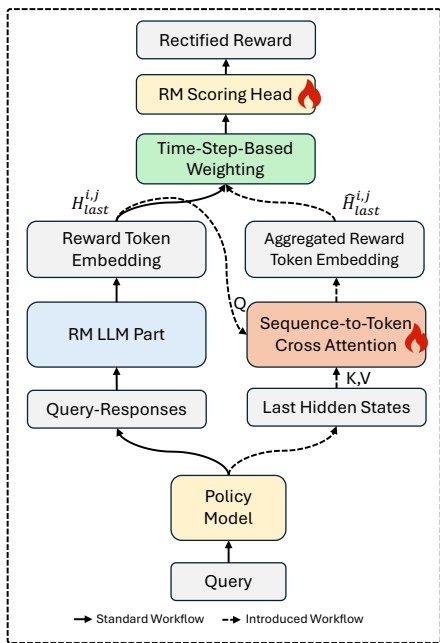

*Figure 4.* The structure of R2M. Building on the dataflow based on solely surface semantic information (left), R2M introduces an additional dataflow based on the policy feedback (right).

we continue to use $h_{i,j}$ to denote the most recent hidden states. This mechanism enables the RM to dynamically capture distribution shifts in real time as the policy evolves.

**Group Reward Entropy Bradley-Terry Loss.** To enhance the robustness of the reward model by incorporating policy feedback during score allocation, we propose the Group Reward Entropy Bradley–Terry (GREBT) Loss. For each query-response group $(x_i, G_i)$, to ensure the reliability of preference labels, we select only the responses with the highest and lowest reward scores to construct the preference pair, resulting in $\{x_i, y_{i,w}, h_{i,w}, y_{i,l}, h_{i,l}\}$. Here, $w$ and $l$ denote winner and loser, respectively, indicating the better and worse options in a preference pair. Then, we can establish

the Bradley-Terry optimization objective as:

$$\mathcal{L}_{\text{BT}}(i;\varphi) = -\log \sigma \big( r_\varphi(x_i, y_{i,w}, h_{i,w}) \\ - r_\varphi(x_i, y_{i,l}, h_{i,l}) \big), \tag{5}$$

which allows the reward model to be continuously optimized as the policy evolves.

However, in practice, the RM often assigns nearly identical scores to responses within a group, especially in the later phases of RL optimization when the responses become more homogeneous. This phenomenon is referred to as the *group degeneration* in RLVR (Yu et al., 2025), and we also observed a similar problem during our R2M training process (discussed in Appendix A.2). To mitigate the impact of the group degeneration, we introduce a entropy regularization term namely Group Reward Entropy to encourage greater reward diversity within each group. Specifically, for each group $(x_i, G_i)$, we first compute the foward pass of the RM $\varphi$ on all samples to get newly allocated reward scores $r_{i,j} = r_\varphi(x_i, y_{i,j}, h_{i,j}), j \in [K]$. We define the Group Reward Entropy (GRE) loss for group $(x_i, G_i)$ as

$$\mathcal{L}_{\text{GRE}}(i;\varphi) = -\sum_{j=1}^{K} p_{i,j} \log p_{i,j}, \text{where} \\ p_{i,j} = \text{softmax}\left( \frac{r_{i,j} - \text{mean}(\mathbf{r})}{\text{std}(\mathbf{r})} \right), \tag{6}$$

where $\mathbf{r} = \{r_{i,1}, r_{i,2}, \dots, r_{i,K}\}$, and $i$ is the group index, the softmax operation is applied across all standardized reward values within the group to get the relative preference of each sample. By minimizing the GRE loss, we minimize the GRE and sharpen the distribution $p_{i,j}$, thereby amplifying the score disparities within the group. Finally, the overall optimization objective of R2M is given by:

$$\mathcal{L}_{\text{GREBT}}(i;\varphi) = (1-\alpha)\mathcal{L}_{\text{BT}}(i;\varphi) + \alpha\mathcal{L}_{\text{GRE}}(i;\varphi), \tag{7}$$

As shown in Theorem 4.1, by guiding the update of $\varphi$, $\mathcal{L}_{\text{GRE}}(i;\varphi;\alpha)$ reduces $\forall C_i$ to a greater degree as the weight coefficient $\alpha \in [0, 1]$ increases.

> **Theorem 4.1.** *(Proof in Appendix A.2) Given $\varphi_\alpha = \arg\min_\varphi \mathcal{L}_{GREBT}(i;\varphi;\alpha)$ and the group degeneration degree $C_i(\varphi)$ for any group $G_i$, we establish the following results: (1) $C_i(\varphi_\alpha) < C_i(\varphi_0)$; (2) $\Delta C_i(\alpha) := C_i(\varphi_0) - C_i(\varphi_\alpha), \forall \alpha_1 < \alpha_2, \Delta C_i(\alpha_1) < \Delta C_i(\alpha_2)$.*

With the GRE loss incorporated into the optimization object, we enable the RM to progressively learn to provide reasonable and more confident reward signals while incorporating real-time policy feedback, thereby allowing it to automatically adapt to the policy's distribution shifts.

**Workflow**. Algorithm 1 illustrates the workflow of our proposed R2M algorithm, The modifications primarily involve utilizing both shallow semantic information $(x_i, y_{i,j})$

and policy feedback $h_{i,j}$ beyond semantics during the Reward Annotation phase, as well as introducing an additional lightweight Reward Model Optimization phase to iteratively update the reward model based on real-time policy feedback.

*Policy Optimization (Lines 10-14).* We retain the same Policy Optimization phase as described in Section 2, with the only difference being that we update the policy feedback for each query-response pair using the real-time updated $\pi_\theta$ as mentioned in Section 4.2.

*Reward Model Optimization (Lines 15-20).* To preserve the general representational capacity of the reward model's LLM part while enhancing the relatively weaker linear projection component, **we solely update the cross-attention component and the scoring head** $\phi$, leaving the LLM part frozen. We discuss detailed motivations in the Appendix H.2. This design significantly reduces the overall computational cost of R2M, ensuring the feasibility of iteratively updating the reward model.

## 5. Experiments and Analyses

In this section, we present the primary experimental results along with their analysis. We set the learning rate of the reward model to $1 \times 10^{-6}$ for the dialogue task and $5 \times 10^{-7}$ for the summarization task, the weight coefficient of the hybrid loss to $\alpha = 0.3$, and the internal width of the cross-attention component to $d = 2048$. During the entire training process, we sample 51.2k trajectories with a maximum length of 512 for the dialogue task, and 1000k trajectories with a maximum length of 50 for the document summarization task.

We integrate R2M into both RLOO and GRPO, and compare them against vanilla RL algorithms. Additionally, we introduce the following three baselines for comparison:

**Pretrained RM**: Built upon the vanilla RL algorithm, we perform full offline pre-training of the vanilla RM using the same queries with golden preference response pairs from UltraFeedback.

**R2M w/o Train**: This variant incorporates policy feedback only during the reward function scoring phase, while keeping the R2M frozen.

**Iterative RM$_{\text{Head}}$**: In each training iteration, we directly compute $\mathcal{L}_{\text{GREBT}}$ using the original reward scores retained in Reward Annotation phase and update the RM's scoring head accordingly.

More experimental details are provided in Appendix G.3, Appendix G.4 and Appendix G.5 due to space constraints.

*Table 1.* AlpacaEval 2 and MT-Bench Results of R2M compared with baselines on Dialogue Tasks. LC and WR denote length-controlled and raw win rate, respectively. Here, **bold** denotes the best performance, underline indicates the second-best performance. Relative changes are compared with the base model (SFT).

| Method | Qwen2.5-3B-Instruct | | | | LLaMA3-8B-Instruct | | | |
| --- | --- | --- | --- | --- | --- | --- | --- | --- |
| | Alpaca-eval | | | MT-Bench | Alpaca-eval | | | MT-Bench |
| | LC(%) | WR(%) | LEN | GPT-4 | LC(%) | WR(%) | LEN | GPT-4 |
| SFT | 15.5 | 15.8 | 2218 | 6.4 | 22.9 | 22.6 | 1899 | 6.9 |
| ReMax | 21.8 (↑ 40.6%) | 25.1 (↑ 58.9%) | 2916 | 6.4 (↑ 0.0%) | 28.7 (↑ 25.3%) | 30.7 (↑ 35.8%) | 2289 | 7.0 (↑ 1.4%) |
| REINFORCE++ | 21.4 (↑ 38.1%) | 26.4 (↑ 67.1%) | 3252 | 6.3 (↓ 1.6%) | 29.3 (↑ 27.9%) | 31.8 (↑ 40.7%) | 2192 | 6.8 (↓ 1.4%) |
| GRPO | 22.7 (↑ 46.5%) | 25.6 (↑ 62.0%) | 3012 | 6.3 (↓ 1.6%) | 29.5 (↑ 28.8%) | 32.6 (↑ 44.2%) | 2216 | 7.0 (↑ 1.4%) |
| + R2M w/o Train | 17.4 (↑ 12.3%) | 19.4 (↑ 22.8%) | 3317 | 6.2 (↓ 3.1%) | 25.6 (↑ 11.8%) | 27.0 (↑ 19.5%) | 2261 | 6.7 (↓ 2.9%) |
| + Pretrained RM | 22.9 (↑ 47.7%) | 28.2 (↑ 78.5%) | 3101 | 6.4 (↑ 0.0%) | 31.5 (↑ 37.5%) | 34.3 (↑ 51.8%) | 2278 | 7.1 (↑ 2.9%) |
| + Iterative RM$_{Head}$ | 23.5 (↑ 51.6%) | 27.8 (↑ 76.0%) | 3050 | 6.5 (↑ 1.6%) | 32.0 (↑ 39.7%) | 33.9 (↑ 50.0%) | 2250 | 7.0 (↑ 1.4%) |
| + R2M | **25.8** (↑ 66.5%) | 30.9 (↑ 95.6%) | 2871 | 6.6 (↑ 3.1%) | **35.6** (↑ 55.4%) | **39.4** (↑ 74.3%) | 2011 | **7.3** (↑ 5.8%) |
| RLOO | 21.9 (↑ 41.3%) | 26.0 (↑ 64.6%) | 3174 | 6.4 (↑ 0.0%) | 28.4 (↑ 24.0%) | 30.2 (↑ 33.6%) | 2186 | 7.1 (↑ 2.9%) |
| + R2M w/o Train | 15.8 (↑ 1.9%) | 20.5 (↑ 29.7%) | 3154 | 6.2 (↓ 3.1%) | 24.4 (↑ 6.5%) | 27.4 (↑ 21.2%) | 2366 | 6.5 (↓ 5.8%) |
| + Pretrained RM | 22.8 (↑ 47.1%) | 27.4 (↑ 73.4%) | 2992 | 6.5 (↑ 1.6%) | 30.5 (↑ 33.2%) | 32.2 (↑ 42.5%) | 2172 | 7.1 (↑ 2.9%) |
| + Iterative RM$_{Head}$ | 23.2 (↑ 50.3%) | 27.0 (↑ 70.9%) | 2950 | 6.5 (↑ 1.6%) | 31.0 (↑ 35.4%) | 31.8 (↑ 40.7%) | 2150 | 7.0 (↑ 1.4%) |
| + R2M | 24.8 (↑ 60.0%) | **31.2** (↑ 97.5%) | 2911 | **6.7** (↑ 4.7%) | 34.5 (↑ 50.7%) | 38.2 (↑ 69.0%) | 2011 | **7.3** (↑ 5.8%) |

## 5.1. Main Results

In this section, we present the experimental results of R2M on dialogue and document summarization tasks. For dialogue task, We considered the current mainstream evaluation frameworks, utilizing queries from UltraFeedback (Cui et al., 2023) for online RL optimization and conducting evaluations with AlpacaEval 2 (Dubois et al., 2024) and MT-Bench (Zheng et al., 2023) , which are widely used chat-based evaluation benchmarks. Next, we considered a classic RLHF task, summarization: given a forum post from Reddit, the policy must generate a summary of the main points in the post.

**(1) R2M consistently achieves superior performance.** As shown in Table 1 and Table 2, the incorporation of policy feedback and iterative updates of the reward model enable R2M to achieve the highest scores across all evaluation metrics. Specifically, both the RLOO+R2M tuned models and the GRPO+R2M tuned models achieve either the best or second best performance across all evaluation metrics. Moreover, they significantly outperform all baseline methods. These results underscore the broad applicability of R2M in preference optimization and its effectiveness in aligning LLMs with human preferences.

Conversely, *R2M w/o Train* not only fails to provide any improvement but actually degrades the performance of vanilla RL algorithms. This indicates that the lighter-weight approach of directly utilizing feedback information without any adaption is not viable.

**(2) R2M enhances the vanilla RM efficiently.** Compared to RLOO, RLOO+R2M achieved a 2.9% to 6.1% increase in LC win rate, a 5.2% to 8.0% increase in raw win rate, and a 6.3% increase in TL;DR win rate. As the sole dif-

*Table 2.* Performance of R2M compared with baselines on Summarization Task (Pythia-2.8B-TL;DR). WR denotes the raw win rate. Relative changes are compared with the base model (SFT).

| Method | WR(%) |
| --- | --- |
| SFT | 42.3 |
| ReMax | 75.1 (↑ 77.5%) |
| REINFORCE++ | 74.3 (↑ 75.6%) |
| GRPO | 75.2 (↑ 77.8%) |
| + R2M w/o Train | 51.1 (↑ 20.8%) |
| + Pretrained RM | 66.3 (↑ 56.7%) |
| + Iterative RM$_{Head}$ | 67.0 (↑ 58.4%) |
| + R2M | 81.0 (↑ 91.5%) |
| RLOO | 75.3 (↑ 78.0%) |
| + R2M w/o Train | 50.6 (↑ 19.6%) |
| + Pretrained RM | 67.3 (↑ 59.1%) |
| + Iterative RM$_{Head}$ | 66.9 (↑ 58.1%) |
| + R2M | **81.6** (↑ 92.9%) |

ference between RLOO+R2M and RLOO lies in the replacement of a frozen RM with one iteratively updated and allocating rewards via policy feedback, these substantial improvements are entirely due to the stronger reward model of RLOO+R2M. This clearly demonstrate the effectiveness of R2M's integration of feedback to iteratively enhance the reward model. To further validate this, we compare the performance of the RM on the test set of UltraFeedback before and after running the R2M+RLOO pipeline, as experimental details shown in Appendix G.6. As shown in Table 3, after iterative updates, R2M achieves accuracy improvements of 5.1% and 6.3% compared to the vanilla RM. These results indicate that R2M significantly enhances the accuracy of the RM, which is crucial for preventing reward overoptimization and improving training effect (Rafailov et al., 2023; Lambert et al., 2024; Adler et al., 2024).

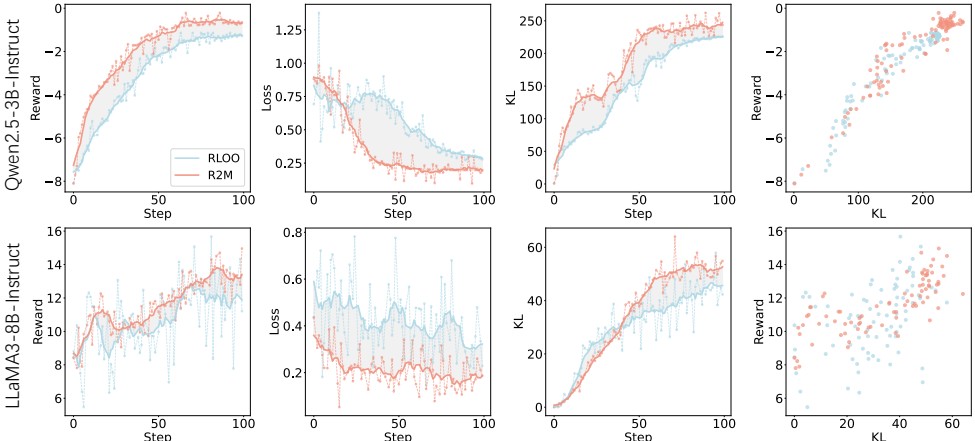

*Figure 5.* We compare RLOO and RLOO+R2M in terms of loss, reward and KL divergence during RL optimization, using Qwen2.5-3B-Instruct and LLaMA3-8B-Instruct as policy models, and Skywork-Reward-V2-Llama-3.1-8B as the reward model. For KL divergence, we calculate it as the average of log probability differences between the reference model and the policy model for each token.

*Table 3.* Comparison of the accuracy of reward models on the test set of UltraFeedback. "Vanilla RM" refers to the frozen reference reward model, while "R2M" represents the reward model before and after the RLOO+R2M pipeline.

| Reward Model | Win Rate(%) | |
|---|---|---|
| | Qwen2.5 | LLaMA3 |
| Vanilla RM | 72.3 | 72.3 |
| R2M (Before-Training) | 68.3 | 69.2 |
| R2M (After-Training) | 77.4 | 78.6 |

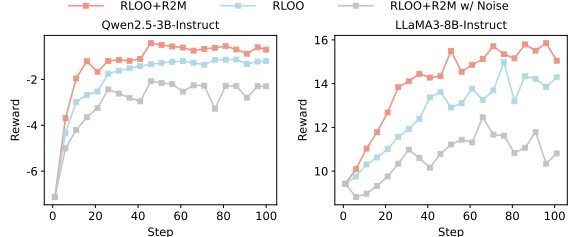

*Figure 6.* Comparison of average rewards in RL Optimization. w/ Noise denotes replacing the feedback in R2M with Gaussian noise.

On the other hand, the performance gain of *Pretrained RM* remains quite limited. This is likely because the vanilla RM has already undergone extensive post-training, causing its capability to approach convergence on the training data. In contrast, **R2M achieves substantial and significant breakthroughs in RM capability with the same amount of training data while introducing far fewer tunable parameters.** Specifically, GRPO + R2M outperforms GRPO + Pretrained RM by 2.9% to 4.1% on Alpaca-Eval LC, and by 2.7% to 5.1% on Alpaca-Eval WR. We can observe a similar phenomenon on the comparison of RLOO+R2M and RLOO+Pretrained RM. This enhancement can be attributed to two factors: real-time alignment with the policy model and additionally introduced deep semantic understanding, thanks to the rich information from policy feedback discussed in Section 3.

**(3) Policy feedback plays a crucial role in R2M updates.**
*Iterative RM$_{Head}$* achieves performance surpassing vanilla RL algorithms and approaching that of the pretrained reward model through lightweight iterative updates to the reward head. This demonstrates the effectiveness of iteratively fine-tuning the reward model. Nevertheless, the improvement over vanilla RL remains quite limited. The primary reason is that **this approach constructs pseudo-labels using reward signals generated by the vanilla RM itself**.

In contrast, R2M exhibits consistent and significant superiority over Iterative RM$_{Head}$ across all experimental settings. This notable performance gain originates from the recomputation of reward signals before RM updates, where policy feedback information is explicitly incorporated into the calculation process. These results strongly indicate that **policy feedback introduces valuable new information into the reconstructed reward distribution**. Furthermore, this supplementary information is effectively leveraged to guide the parameter updates of R2M via our tailored $\mathcal{L}_{GREBT}$.

### 5.2. Analysis

In this section, we present additional analytical experiments to clarify the reasons behind R2M's effectiveness in RL optimization from a principled perspective.

**(1) R2M maintains reward consistency while allocating higher rewards.** Every 5 training steps, we sampled 128 queries from the *test set*, prompted $\pi_\theta$ to generate responses. Then, we scored them with the reward model and illustrated the average results in Figure 6. The iteratively updated reward model in RLOO+R2M exhibits a similar reward trend compared to the vanilla frozen RM in RLOO and

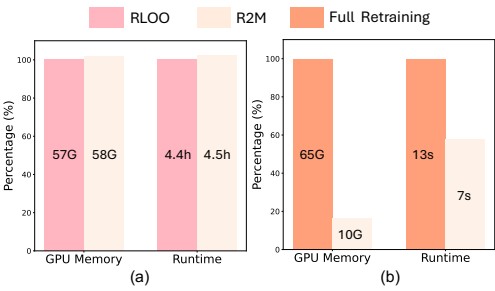

*Figure 7.* (a) Computational cost comparison between RLOO and RLOO+R2M. (b) Computational cost comparison between full reward model updates and lightweight updates in R2M.

consistently assigns higher rewards, indicating that R2M can reliably provide reasonable and well-calibrated reward signals. In contrast, reward scores from R2M w/ Noise are significantly lower, which confirms that policy feedback carries beneficial information for enhancing the vanilla reward model, consistent with Section 3. We hypothesize that the higher reward allocation results of R2M stems from the GRE loss, which encourages the RM to assign higher reward values to high-quality responses with greater confidence.

**(2) R2M encourages substantial and effective policy updates.** Figure 5 illustrates the training dynamics for the dialogue task. RLOO+R2M demonstrates a significantly higher reward curve and lower loss curve compared to RLOO. Generally, this indicates more effective training outcomes. From the perspective of KL divergence, R2M encourages larger parameter shifts in the model to achieve greater rewards. Furthermore, RLOO+R2M yields a noticeably denser concentration of points in the high-KL-divergence & high-reward region compared to vanilla RLOO. This indicates that **R2M effectively encourages more aggressive policy updates by assigning systematically higher rewards**. Aggressive policy updates readily lead to reward overoptimization (Coste et al., 2023), but R2M still outperform the vanilla RL algorithms significantly demonstrated in Section 5.1. In summary, R2M effectively improves the RM's resistance to policy's exploitation of specific patterns, enabling more aggressive policy updates in the correct direction without triggering reward overoptimization.

### 5.3. Computational Cost Analysis

R2M is lightweight and compute-efficient. In Figure 7(a), we compare the peak single-GPU memory footprint and total runtime of RLOO+R2M against RLOO in the LLaMA environment. In Figure 7(b), we compare the peak GPU memory consumption and runtime between performing a full reward model update in a single training iteration and the lightweight update mechanism of R2M. To avoid out-of-memory (OOM) issues and isolate the cost of RM updates, we disable gradient computation on the policy model.

**R2M substantially reduces the time and memory over-**

**head of full reward model updates**, while incurring negligible additional computational cost compared to the significant performance improvements it achieves. This can attribute to two main factors. First, policy feedback can be directly obtained and its aggregation solely involves lightweight attention computations. Second, R2M does not update the reward model's LLM part, and its cross-attention module and scoring head are relatively lightweight.

### 5.4. Ablation Study

In this section, we perform detailed ablation studies to assess the effectiveness of each component in R2M. Based on the LLaMA3 setup outlined in Section 5.1, we systematically remove key modules of R2M and evaluate their impact on experimental results, as presented in Table 4.

*Table 4.* Ablation study results on AlpacaEval 2.

| Method | LC(%) | WR(%) | LEN |
|---|---|---|---|
| SFT | 22.9 | 22.6 | 1899 |
| RLOO | 28.4 (↓ 17.7%) | 30.2 (↓ 20.9%) | 2186 |
| +R2M w/ Noise | 25.4 (↓ 26.4%) | 26.4 (↓ 30.9%) | 2276 |
| +R2M w/o Train | 24.4 (↓ 29.3%) | 27.4 (↓ 28.3%) | 2366 |
| +R2M w/o BT | 31.5 (↓ 8.7%) | 35.7 (↓ 6.5%) | 2116 |
| +R2M w/o GRE | 32.3 (↓ 6.4%) | 36.2 (↓ 5.2%) | 2191 |
| +R2M | 34.5 | 38.2 | 2011 |

**R2M w/ Noise & R2M w/o Train**: For R2M w/ Noise, we replace the feedback information with Gaussian noise of equivalent mean and variance. We observe that both methods yield only similar and limited performance improvements on the SFT model, and these gains are substantially smaller than those achieved by standard R2M. The improvement primarily stems from the dominant role of the original RTE in the early stage of training. Apart from this, the similarity in experimental results suggests that injecting policy feedback into an unfine-tuned reward model yields effects essentially equivalent to injecting noise. These results suggest that to effectively incorporate feedback information, updating R2M is necessary, which aligns with Section 5.1.

**R2M w/o BT & R2M w/o GRE**: We compare the results of R2M trained with the GREBT loss against those of R2M optimized with a single objective (i.e., either the GRE loss or the BT loss alone). Compared to R2M trained with GREBT loss, we observed that removing the BT loss resulted in a decrease of 3.0 and 2.5 in LC and WR scores, respectively. This phenomenon stems from the inherent reliance of RM training on the BT loss. On the other hand, when the GRE loss was removed, the scores dropped 2.2 and 2.0 respectively. This is mainly caused by the group degeneration phenomenon mentioned in Section 4.2. These results clearly indicate that utilizing a mixed loss as the

optimization objective outperforms a single objective.

In summary, each component of R2M is indispensable and effective, as the ablation of any single component leads to a significant performance degradation.

## 6. Discussion

### 6.1. Potential in Multi-node Communication Scenarios

R2M demonstrates strong scalability in multi-node communication scenarios. Although cross-node communication is a valid concern for distributed deployment, this overhead can be substantially reduced through the following design choices.

**Communication Volume Reduction.** Rather than transmitting the full hidden state tensor $(B \times S \times D_p)$, we can place a lightweight copy of the cross-attention module on the policy node (parameter count: only $2 \times (D_r \times d + D_p \times d)$). During rollout, only the RM's reward token embedding $(B \times D_p)$ is sent to the policy node for cross-attention aggregation, and the updated embedding is sent back. This reduces per-step communication from $B \times S \times D_p$ to $2 \times B \times D_p$. Critically, this cost no longer scales with sequence length $S$.

**Asynchronous Updates.** Since the RM update is faster than policy optimization, the updated cross-attention module can be synchronized to the policy node while the policy continues training, avoiding network blocking.

With these optimizations, R2M adds only $2 \times B \times D_p$ in data transfer during rollout, and training-time communication remains identical to standard RL.

### 6.2. The Effectiveness of GRE Loss

When trajectory qualities are similar and reward noise exceeds the true score differences, GRPO may reinforce some trajectories randomly while suppressing others. The GRE loss addresses this by **enlarging reward gaps**, allowing the policy to preferentially reinforce trajectories with slight advantages rather than updating arbitrarily.

**No Artificial Inflation.** When all responses are truly identical, **the GRE loss gradient is exactly zero**, ensuring that it does not create artificial distinctions. GRE activates only when genuine quality differences exist, sharpening the reward distribution without altering the correct ranking, thus alleviating *group degeneration*.

Formally, if all standardized inputs $z_j$ are equal, Softmax produces a uniform distribution $p_j = 1/K$, and the GRE loss (negative entropy) reaches its maximum $\log K$. The gradient w.r.t. $z_i$ is:

$$\frac{\partial \mathcal{L}_{\text{GRE}}}{\partial z_i} = \sum_{j=1}^{K} \frac{\partial \mathcal{L}_{\text{GRE}}}{\partial p_j} \frac{\partial p_j}{\partial z_i}$$

$$= (\log K - 1) \sum_{j=1}^{K} p_j(\delta_{ij} - p_i)$$

$$= 0$$

implying no parameter updates occur. Thus, GRE only amplifies differences once meaningful quality distinctions exist.

**Robustness under Reward Noise.** For responses with identical ground truth rewards, observed rewards may differ due to noise:

$$r_j = r_j^* + \epsilon_j, \quad r_k = r_k^* + \epsilon_k, \quad \epsilon_j, \epsilon_k \sim \mathcal{N}(0, \sigma^2).$$

Since GRE operates over the expected distribution rather than individual samples, the expected gradient satisfies:

$$\mathbb{E}\left[\frac{\partial \mathcal{L}_{\text{GRE}}}{\partial \Delta r_{jk}}\right] = 0 \quad \text{when } r_j^* = r_k^*,$$

ensuring that spurious noise does not systematically bias the model.

When true rewards differ ($r_j^* \neq r_k^*$), this symmetry breaks, yielding non-zero expected gradients in the correct direction. Consequently, GRE **robustly amplifies genuine quality differences** while washing out noise-induced variations over the data distribution.

## 7. Conclusion

To achieve real-time alignment towards policy's distribution shifts efficiently, we propose **R2M**, a novel lightweight RLHF framework. By incorporating the policy's evolving hidden states, R2M enhances the vanilla RM while maintaining robustness against reward overoptimization. Without modifying current RLHF algorithms, simply integrating R2M into the framework achieves significant performance improvements while introducing only marginal additional computational costs.

## Impact Statement

Our proposed R2M offers several significant advantages and has far-reaching potential applications. By incorporating real-time feedback from the policy model, R2M addresses a critical limitation of traditional reward models, enabling iterative alignment with the policy model and more accurate reward allocation. Its seamless integration with current RLHF algorithms without altering the core mechanism and minimal computational overhead make it highly practical for both research and real-world use. In natural language processing (NLP), R2M can enhance chatbots, virtual assistants, and content generation systems, improving user experiences and text quality.

While our method has broad applicability across domains, we do not foresee specific societal risks or negative impacts that require special consideration, as R2M focuses on enhancing the reward model in RL optimization of RLHF framework and maintains the ethical and societal implications consistent with standard RLHF practices.

## Acknowledgements

This research was supported by NSFC (No. 62276015, No. 62506024, No. 62506319) and GW2025-09.

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

# A. Theoretical Analysis

This section provides theoretical support for the core components of R2M: the incorporation of the internal hidden states from policy models and the GRE loss. These analyses theoretically justify why R2M can enhance the vanilla RM while maintaining robustness against reward overoptimization.

## A.1. Proof of Theorem 3.1

**Restatement of Theorem 3.1:** Introducing policy hidden states into the vanilla reward model strictly tightens the upper bound on reward misalignment compared to the vanilla reward model, when the post-fusion alignment quality $\gamma^{(t)} > 0$:

$$\epsilon_{\text{R2M}}^{(t)} \leq (1 - \gamma^{(t)})^{1/2} \cdot C + \Delta\mathcal{D}^{(t)} \cdot L, \tag{8}$$

$$\epsilon_{\text{vanilla}}^{(t)} \leq C + \Delta\mathcal{D}^{(t)} \cdot L, \tag{9}$$

where $C = L_h \cdot D \cdot \sqrt{2}$ represents the worst-case semantic deviation bound in the fused representation space.

Let $\pi_\theta^{(t)}$ denote the policy model at RL training step $t$. Classically, the reward model $r_\varphi$ and the policy model $\pi_\theta$ shares the same initial distribution $\mathcal{D}^{(0)} \sim \pi_\theta^{(0)}$ (Ouyang et al., 2022a; Ahmadian et al., 2024). As training proceeds, the policy induces a drifted distribution $\mathcal{D}^{(t)}$, causing reward misalignment in models that do not adapt to this shift.

R2M considers a more general scenario, where **the reward model and the policy model are heterogeneous**. In this case, the aforementioned issues persist as well, and the semantic gap stems not only from distribution shift but also from the profound discrepancies between the foundation models. We train a lightweight **Sequence-to-Token Cross Attention** module $M_{\text{ca}}$ (as introduced in Section 4.1) to bridge this gap: it takes policy hidden states as key/value and reward model's internal features as query, producing post-fusion features $h_f^{(t)} = M_{\text{ca}}(h^{(t)})$ in the reward model's representation space.

The following definitions apply to the fused representation space:

**Definition A.1** (Distribution Shift Degree). $\Delta\mathcal{D}^{(t)} = \text{TV}(\mathcal{D}^{(t)}\|\mathcal{D}^{(0)})$.

**Definition A.2** (Reward Misalignment Error). $\epsilon^{(t)} = \mathbb{E}_{(x,y)\sim\mathcal{D}^{(t)}}\big|r_\phi(x,y) - r^*(x,y)\big|$, where $r^*(x,y)$ is the true underlying human preference reward.

**Definition A.3** (Post-Fusion Hidden State Alignment Quality). $\gamma^{(t)} = \mathbb{E}_{(x,y)\sim\mathcal{D}^{(t)}} \cos\big(h_f^{(t)}(x,y),\, h_f^*(x,y)\big) \in [0, 1]$, where $h_f^{(t)}(x,y)$ is the fused hidden state (direct $h^{(t)}$ or $M_{\text{ca}}(h^{(t)})$), and $h_f^*(x,y)$ is the ideal fused representation corresponding to the truly preferred response.

**Definition A.4** (Lipschitz Constants). $L$ is the Lipschitz constant of the reward head w.r.t. the query-response pair; $L_h$ is the Lipschitz constant w.r.t. the fused hidden state.

**Definition A.5** (Hidden State Norm Bound). $\|h_f\|_2 \leq D$ and $\|h_f^*\|_2 \leq D$ for all fused hidden states.

**Definition A.6** (Maximum Hidden-State Semantic Deviation). $C := L_h \cdot D \cdot \sqrt{2}$ is the worst-case reward deviation in the fused representation space caused by the most misaligned fused hidden state (achieved in the limit as $\cos(h_f, h_f^*) \to -1$).

This constant is the prefactor that arises from bounding the Euclidean distance between two vectors of bounded norm $\|h_f\|_2 \leq D$ and $\|h_f^*\|_2 \leq D$. Specifically, by the law of cosines, we have

$$\|h_f - h_f^*\|_2^2 = \|h_f\|_2^2 + \|h_f^*\|_2^2 - 2\|h_f\|_2\|h_f^*\|_2 \cos(h_f, h_f^*) \leq 2D^2(1 - \cos(h_f, h_f^*)). \tag{10}$$

Taking the square root gives the tight worst-case distance bound

$$\|h_f - h_f^*\|_2 \leq D\sqrt{2(1 - \cos(h_f, h_f^*))}. \tag{11}$$

Since the reward head is $L_h$-Lipschitz continuous with respect to the fused hidden state, the induced deviation in reward is bounded by

$$\big|r_\phi(x,y,h_f) - r_\phi(x,y,h_f^*)\big| \leq L_h \cdot \|h_f - h_f^*\|_2 \leq L_h \cdot D \cdot \sqrt{2(1 - \cos(h_f, h_f^*))}. \tag{12}$$

When $\cos(h_f, h_f^*) \to -1$ (the most adverse case), this reaches its maximum value of $L_h \cdot D \cdot \sqrt{4} = 2L_hD$. For general alignment quality $\gamma = \cos(h_f, h_f^*)$, we can factor the expression as

$$L_h \cdot D \cdot \sqrt{2(1-\gamma)} = (1-\gamma)^{1/2} \cdot (L_h \cdot D \cdot \sqrt{2}), \tag{13}$$

which is exactly the form used in the proof: the misalignment term is at most $(1 - \gamma^{(t)})^{1/2} \cdot C$. Thus $C = L_h \cdot D \cdot \sqrt{2}$ conveniently encapsulates the geometric worst-case factor from the bounded-norm ball in the fused representation space.

*Proof.* We separately bound the misalignment error for the vanilla reward model and for R2M (with hidden state fusion), then compare the resulting upper bounds. All quantities are defined in the fused representation space; the vanilla model is treated as having no access to fused hidden state information, corresponding to the worst-case scenario in this space.

**Part 1: Upper bound for vanilla reward model $\epsilon_{\text{vanilla}}^{(t)}$** The vanilla reward model $r_{\phi,\text{vanilla}}(x, y)$ receives only the query-response pair. We bound its error via the ideal reward $r^*(x, y)$ and an auxiliary quantity $r_\phi(x, y, h_f^*)$ (the output of the R2M architecture when provided with the ideal fused hidden state $h_f^*$):

$$\left| r_{\phi,\text{vanilla}}(x, y) - r^*(x, y) \right| \le \left| r_{\phi,\text{vanilla}}(x, y) - r_\phi(x, y, h_f^*) \right| + \left| r_\phi(x, y, h_f^*) - r^*(x, y) \right|. \tag{14}$$

The second term is bounded using the training distribution shift and Lipschitz continuity w.r.t. $(x, y)$:

$$\mathbb{E}_{\mathcal{D}^{(t)}} \left| r_\phi(x, y, h_f^*) - r^*(x, y) \right| \le \Delta\mathcal{D}^{(t)} \cdot L. \tag{15}$$

For the first term, note that the vanilla model has no mechanism to incorporate policy-specific hidden state information. In the worst case, its output deviates from the ideal fused-R2M output $r_\phi(x, y, h_f^*)$ by up to the maximum semantic deviation inducible in the fused representation space (as defined by $C$). This holds under the assumption that the vanilla model's representation capacity does not exceed the range spanned by the fused space under ideal alignment, yielding:

$$\left| r_{\phi,\text{vanilla}}(x, y) - r_\phi(x, y, h_f^*) \right| \le C. \tag{16}$$

Taking expectation over $\mathcal{D}^{(t)}$ gives

$$\epsilon_{\text{vanilla}}^{(t)} \le C + \Delta\mathcal{D}^{(t)} \cdot L. \tag{17}$$

**Part 2: Upper bound for R2M $\epsilon_{\text{R2M}}^{(t)}$** For the hidden-state-aware reward model $r_\phi(x, y, h_f)$ (where $h_f$ is obtained directly or via the trained $M_{\text{ca}}$), we decompose:

$$\left| r_\phi(x, y, h_f) - r^*(x, y) \right| \le \left| r_\phi(x, y, h_f) - r_\phi(x, y, h_f^*) \right| + \left| r_\phi(x, y, h_f^*) - r^*(x, y) \right|. \tag{18}$$

The second term is bounded by (15). For the first term, by $L_h$-Lipschitz continuity w.r.t. the fused hidden state:

$$\left| r_\phi(x, y, h_f) - r_\phi(x, y, h_f^*) \right| \le L_h \cdot \|h_f - h_f^*\|_2. \tag{19}$$

From the cosine similarity definition and norm bounds $\|h_f\|_2, \|h_f^*\|_2 \le D$, we obtain (in the worst case):

$$\|h_f - h_f^*\|_2^2 \le 2D^2(1 - \cos(h_f, h_f^*)), \tag{20}$$

$$\|h_f - h_f^*\|_2 \le D\sqrt{2(1 - \gamma^{(t)})}. \tag{21}$$

Substituting yields

$$\left| r_\phi(x, y, h_f) - r_\phi(x, y, h_f^*) \right| \le L_h \cdot D \cdot \sqrt{2(1 - \gamma^{(t)})} = (1 - \gamma^{(t)})^{1/2} \cdot C. \tag{22}$$

Taking expectation over $\mathcal{D}^{(t)}$ finally gives

$$\epsilon_{\text{R2M}}^{(t)} \le (1 - \gamma^{(t)})^{1/2} \cdot C + \Delta\mathcal{D}^{(t)} \cdot L. \tag{23}$$

**Comparison** The trained cross-attention module (or direct fusion) enables non-trivial post-fusion alignment ($\gamma^{(t)} \in (0, 1]$ in practice). Thus $(1 - \gamma^{(t)})^{1/2} < 1$, implying

$$(1 - \gamma^{(t)})^{1/2} \cdot C + \Delta\mathcal{D}^{(t)} \cdot L < C + \Delta\mathcal{D}^{(t)} \cdot L. \tag{24}$$

This establishes that R2M enjoys a strictly tighter upper bound on reward misalignment whenever $\gamma^{(t)} > 0$. $\qquad\square$

**Corollary A.7.** *When hidden states are perfectly aligned after fusion ($\gamma^{(t)} = 1$), the upper bound simplifies to $\epsilon_{R2M}^{(t)} \le \Delta\mathcal{D}^{(t)} \cdot L$, i.e., misalignment is solely controlled by distribution shift.*

**Corollary A.8.** *The benefit of hidden state fusion becomes more pronounced as distribution shift $\Delta\mathcal{D}^{(t)}$ grows, because the reducible portion $\left[ 1 - (1 - \gamma^{(t)})^{1/2} \right] C$ constitutes a larger relative improvement in the total bound. This provides theoretical support for the observed long-horizon stability of R2M in experiments, even under significant distribution drift.*

In summary, fusing policy hidden states provably compresses the upper bound of reward misalignment by leveraging post-fusion alignment quality $\gamma^{(t)}$. This offers theoretical grounding for mitigating reward misalignment via incorporating policy feedback into the reward models.

### A.2. Proof of Theorem 4.1

**Restatement of Theorem 4.1:** The Group Reward Entropy (GRE) term in the GREBT loss strictly mitigates group degeneration of the reward model, and the mitigation strength increases monotonically with the weighting coefficient $\alpha \in (0, 1]$. For any fixed group $G_i$, let $\varphi_0 = \arg\min_\varphi \mathcal{L}_{\text{BT}}(i; \varphi)$ denote the minimizer of the pure Bradley-Terry (BT) loss, and $\varphi_\alpha = \arg\min_\varphi \mathcal{L}_{\text{GREBT}}(i; \varphi; \alpha) = \arg\min_\varphi \left[ (1 - \alpha)\mathcal{L}_{\text{BT}}(i; \varphi) + \alpha\mathcal{L}_{\text{GRE}}(i; \varphi) \right]$ denote the minimizer of the GREBT loss. If the reward model exhibits group degeneration at $\varphi_0$, where $C_i(\varphi_0) > 0$ and $C_i(\varphi)$ denotes the group degeneration degree, then: (1) $C_i(\varphi_\alpha) < C_i(\varphi_0)$ ; (2) $\Delta C_i(\alpha) := C_i(\varphi_0) - C_i(\varphi_\alpha)$ is strictly increasing in $\alpha$.

We first formalize key variables and definitions for a preference group with $K$ responses, then proceed with the proof by verifying the two claims sequentially. All quantities are defined for a fixed preference group $i$, and we omit the subscript $i$ for notational simplicity where no ambiguity arises.

**Definition A.9** (Reward Score and Statistic). For a preference group $(x, G)$ with $K$ responses $\{y_j\}_{j=1}^K$, $r_j = r_\varphi(x, y_j, h_j)$ is the reward score assigned by the reward model with parameter $\varphi$; $\mu = \frac{1}{K} \sum_{j=1}^K r_j$ is the mean reward score, and $\sigma = \sqrt{\frac{1}{K} \sum_{j=1}^K (r_j - \mu)^2} > 0$ is the standard deviation of reward scores in the group.

**Definition A.10** (Standardized Reward and Softmax Probability). $z_j = \frac{r_j - \mu}{\sigma}$ is the standardized reward score (normalized to zero mean and unit variance); $p_j = \frac{\exp(z_j)}{\sum_{k=1}^K \exp(z_k)}$ is the softmax-normalized probability of the $j$-th response over the group, with $\sum_{j=1}^K p_j = 1$.

**Definition A.11** (Group Degeneration Degree). $C(\varphi) \triangleq \mathcal{L}_{\text{GRE}}(\varphi) = -\sum_{j=1}^K p_j \log p_j$ is the group degeneration degree, equivalent to the GRE loss. It takes values in $[0, \log K]$, where:

- $C(\varphi) = \log K$ (maximum entropy) implies complete group degeneration ($\sigma \to 0^+$), with the model assigning nearly identical rewards to all responses ($p_j = 1/K$ for all $j$);

- $C(\varphi) \to 0$ (minimum entropy) implies no group degeneration, with one response dominating the reward distribution ($p_j \to 1$ for a single $j$, $p_k \to 0$ for $k \ne j$).

**Definition A.12** (GREBT Loss). The fused Group Reward Entropy Bradley-Terry (GREBT) loss is a weighted combination of the pure BT loss and the GRE loss:

$$\mathcal{L}_{\text{GREBT}}(\varphi, \alpha) = (1 - \alpha)\mathcal{L}_{\text{BT}}(\varphi) + \alpha C(\varphi), \tag{25}$$

where $\alpha \in (0, 1]$ is the weighting coefficient that controls the strength of the group degeneration mitigation.

All derivations hold under standard regularity conditions for optimization:

**Assumption A.13.** $\mathcal{L}_{\text{BT}}(\varphi)$ and $C(\varphi)$ are continuously differentiable in $\varphi$.

**Assumption A.14.** the Hessian of $\mathcal{L}_{\text{GREBT}}(\varphi, \alpha)$ is positive definite at the minimizer $\varphi_\alpha$, ensuring the uniqueness of the minimizer and valid application of the implicit function theorem.

*Proof.* We prove the two claims of the theorem in two parts: Part 1 verifies the strict mitigation of group degeneration, and Part 2 proves the monotonic increase of the mitigation strength with $\alpha$.

**Part 1: Strict Mitigation of Group Degeneration**    Since $\varphi_0$ is the minimizer of the pure BT loss, for any parameter $\varphi$, we have the fundamental inequality:

$$\mathcal{L}_{\text{BT}}(\varphi) \geq \mathcal{L}_{\text{BT}}(\varphi_0). \tag{26}$$

For the GREBT minimizer $\varphi_\alpha$, this implies

$$\mathcal{L}_{\text{BT}}(\varphi_\alpha) \geq \mathcal{L}_{\text{BT}}(\varphi_0). \tag{27}$$

Because $\varphi_\alpha$ minimizes the GREBT loss, it must satisfy the optimality condition relative to $\varphi_0$:

$$(1 - \alpha)\mathcal{L}_{\text{BT}}(\varphi_\alpha) + \alpha C(\varphi_\alpha) \leq (1 - \alpha)\mathcal{L}_{\text{BT}}(\varphi_0) + \alpha C(\varphi_0). \tag{28}$$

Rearranging terms to isolate the differences in BT loss and degeneration degree gives:

$$(1 - \alpha)\big(\mathcal{L}_{\text{BT}}(\varphi_\alpha) - \mathcal{L}_{\text{BT}}(\varphi_0)\big) \leq \alpha\big(C(\varphi_0) - C(\varphi_\alpha)\big). \tag{29}$$

From the above inequality, as $1 - \alpha \geq 0$ and a nonnegative loss difference, the left-hand side is nonnegative. Since $\alpha > 0$, the right-hand side must also be nonnegative, which immediately implies

$$C(\varphi_\alpha) \leq C(\varphi_0). \tag{30}$$

We now rule out the equality case $C(\varphi_\alpha) = C(\varphi_0)$. If equality held, the right-hand side of the above inequality would be zero, forcing the left-hand side to also be zero , i.e., $\mathcal{L}_{\text{BT}}(\varphi_\alpha) = \mathcal{L}_{\text{BT}}(\varphi_0)$). This would mean $\varphi_0$ is also a minimizer of the GREBT loss, which contradicts the group degeneration assumption $C(\varphi_0) > 0$: in the degeneration regime ($\sigma \approx 0$), the BT loss landscape is nearly flat ($\nabla_\varphi \mathcal{L}_{\text{BT}}(\varphi_0) \approx 0$), and small perturbations to $\varphi$ that increase reward polarization (raise $\sigma$) incur a negligible increase or even decrease in $\mathcal{L}_{\text{BT}}$, while causing a substantial decrease in $C(\varphi)$ (from near $\log K$ to lower entropy values).

Since $C(\varphi)$ is continuously differentiable, there exists a strict descent direction for the GREBT loss at $\varphi_0$ that reduces $C(\varphi)$ without a compensating increase in $\mathcal{L}_{\text{BT}}$. Thus $\varphi_0$ cannot be a minimizer of the GREBT loss, and equality $C(\varphi_\alpha) = C(\varphi_0)$ is impossible. We conclude

$$C(\varphi_\alpha) < C(\varphi_0). \tag{31}$$

**Part 2: Monotonic Increase of Mitigation Strength with $\alpha$**    We first establish that the group degeneration degree $C(\varphi_\alpha)$ is strictly decreasing in $\alpha$; the strict monotonicity of $\Delta C_i(\alpha)$ follows directly from this result.

The GREBT minimizer $\varphi_\alpha$ satisfies the first-order optimality condition:

$$(1 - \alpha)\nabla_\varphi \mathcal{L}_{\text{BT}}(\varphi_\alpha) + \alpha \nabla_\varphi C(\varphi_\alpha) = 0. \tag{32}$$

Rearranging the above equation gives an explicit relation between the gradients of the BT loss and degeneration degree:

$$\nabla_\varphi \mathcal{L}_{\text{BT}}(\varphi_\alpha) = -\frac{\alpha}{1 - \alpha}\nabla_\varphi C(\varphi_\alpha). \tag{33}$$

We differentiate both sides of the first-order optimality condition with respect to $\alpha$, applying the product rule and chain rule for differentiation. For a differentiable function $f(\varphi(\alpha))$, its derivative w.r.t. $\alpha$ is $\nabla_\varphi f(\varphi(\alpha))^\top \frac{\partial \varphi}{\partial \alpha}$; this gives:

$$-\nabla_\varphi \mathcal{L}_{\text{BT}}(\varphi_\alpha) + (1-\alpha)\nabla_\varphi^2 \mathcal{L}_{\text{BT}}(\varphi_\alpha)\frac{\partial \varphi_\alpha}{\partial \alpha} + \nabla_\varphi C(\varphi_\alpha) + \alpha \nabla_\varphi^2 C(\varphi_\alpha)\frac{\partial \varphi_\alpha}{\partial \alpha} = 0. \tag{34}$$

Substitute the gradient relation into the above equation to eliminate $\nabla_\varphi \mathcal{L}_{\text{BT}}(\varphi_\alpha)$:

$$\frac{\alpha}{1-\alpha}\nabla_\varphi C(\varphi_\alpha) + \nabla_\varphi C(\varphi_\alpha) + \underbrace{\left[(1-\alpha)\nabla_\varphi^2 \mathcal{L}_{\text{BT}}(\varphi_\alpha) + \alpha \nabla_\varphi^2 C(\varphi_\alpha)\right]}_{\nabla_\varphi^2 \mathcal{L}_{\text{GREBT}}(\varphi_\alpha)}\frac{\partial \varphi_\alpha}{\partial \alpha} = 0. \tag{35}$$

Simplify the gradient terms and denote the Hessian of the GREBT loss as $H(\alpha) = \nabla_\varphi^2 \mathcal{L}_{\text{GREBT}}(\varphi_\alpha)$ (positive definite by non-degeneracy assumption):

$$\frac{1}{1-\alpha}\nabla_\varphi C(\varphi_\alpha) + H(\alpha)\frac{\partial \varphi_\alpha}{\partial \alpha} = 0. \tag{36}$$

Solving for the derivative of the minimizer w.r.t. $\alpha$ yields:

$$\frac{\partial \varphi_\alpha}{\partial \alpha} = -\frac{1}{1-\alpha}H(\alpha)^{-1}\nabla_\varphi C(\varphi_\alpha). \tag{37}$$

We now compute the derivative of the group degeneration degree $C(\varphi_\alpha)$ w.r.t. $\alpha$, again applying the chain rule:

$$\frac{dC(\varphi_\alpha)}{d\alpha} = \nabla_\varphi C(\varphi_\alpha)^\top \frac{\partial \varphi_\alpha}{\partial \alpha}. \tag{38}$$

Substitute the derivative of the minimizer into the above equation to obtain the final expression for the derivative:

$$\frac{dC(\varphi_\alpha)}{d\alpha} = -\frac{1}{1-\alpha}\nabla_\varphi C(\varphi_\alpha)^\top H(\alpha)^{-1}\nabla_\varphi C(\varphi_\alpha). \tag{39}$$

The right-hand side of the above equation is strictly negative for all $\alpha \in (0,1]$: since $\alpha < 1$, $\frac{1}{1-\alpha} > 0$; $H(\alpha)^{-1}$ is positive definite as the inverse of a positive definite matrix $H(\alpha)$; $\nabla_\varphi C(\varphi_\alpha) \neq 0$, as the model is in the degeneration regime $C(\varphi_0) > 0$, and $\varphi_\alpha$ is not the maximum entropy point where the gradient of $C(\varphi)$ vanishes.

A positive definite quadratic form $\nabla^\top H^{-1}\nabla$ is strictly positive for non-zero $\nabla$, so we conclude:

$$\frac{dC(\varphi_\alpha)}{d\alpha} < 0. \tag{40}$$

This means $C(\varphi_\alpha)$ is a strictly decreasing function of $\alpha$. The degeneration reduction is defined as $\Delta C(\alpha) = C(\varphi_0) - C(\varphi_\alpha)$, with $C(\varphi_0)$ a constant (independent of $\alpha$). The derivative of the reduction is

$$\frac{d\Delta C(\alpha)}{d\alpha} = -\frac{dC(\varphi_\alpha)}{d\alpha} > 0, \tag{41}$$

which implies $\Delta C(\alpha)$ is strictly increasing in $\alpha \in (0,1]$.

$\square$

**Corollary A.15.** *When the weighting coefficient $\alpha = 1$, GREBT loss degrades to pure GRE loss, the group degeneration degree is minimized (i.e., $C(\varphi_1) = \min_\varphi C(\varphi)$), and the degeneration reduction $\Delta C(1)$ achieves its maximum value. This corresponds to the strongest mitigation of group degeneration by the GRE term.*

**Corollary A.16.** *As $\alpha \to 0^+$, the GREBT loss converges to the pure BT loss, and the degeneration reduction $\Delta C(\alpha) \to 0$, i.e., no mitigation of group degeneration. This recovers the vanilla BT loss regime as a limiting case of the GREBT loss.*

In summary, group degeneration occurs in the late stage of R2M training. As noted in Section 4.2, under the guidance of the same RM and with feedback information from the identical policy model, R2M suffers from more severe group degeneration. However, the GRE term in the GREBT loss provably induces a strict reduction in group degeneration, with the mitigation strength tunable via the weighting coefficient $\alpha$. The monotonicity of the reduction with $\alpha$ provides a theoretical guarantee for adjusting the trade-off between preference ranking (BT loss) and group degeneration mitigation (GRE loss) in our iterative reward model optimization.

# B. Related Work

**REINFORCE-based RLHF Algorithms.**   RLHF is a critical technique for aligning large language models with human preferences (Ouyang et al., 2022b; He et al., 2025; Bai et al., 2022a; Zhang et al., 2023). The classical RLHF pipeline typically comprises three phases: supervised fine-tuning (Geng et al., 2023; Zou et al., 2025; Chen et al., 2025b), reward model training (Gao et al., 2023), and policy optimization against the reward model (Schulman et al., 2017). As a classic reinforcement learning algorithm, Proximal Policy Optimization (PPO) (Schulman et al., 2017) is widely used in the third stage of RLHF. Recently, many researchers have proposed a series of REINFORCE-based methods, such as ReMax (Li et al., 2023), RLOO (Ahmadian et al., 2024), GRPO (Shao et al., 2024) and REINFORCE++ (Hu, 2025) to avoid the computational overhead associated with the critic model while still obtaining relatively accurate sequence-wise advantage estimations. These methods design alternative techniques to calculate the baseline reward for each prompt as the advantage estimation. (Yang et al., 2026) provides a principled theoretical analysis of group-based advantage estimation. Subsequent algorithm variants have continued to emerge and provided multifaceted optimizations for them (Huang et al., 2025; Zhang et al., 2026; Huang et al., 2026). In addition to the methods discussed above, a wide range of advanced techniques have been proposed in recent years to address various challenges in representation learning, model optimization, reasoning, and generative modeling. These include progress in interpretable representation learning (Li et al., 2025a), prompt-based structural modeling (Li et al., 2025c), diffusion-driven restoration (Li et al., 2025b), efficient transformer architectures for visual modeling (Fu et al., 2022), prompt-guided sequence modeling (Cai et al., 2023; 2024), parameter-efficient tuning strategies (Cai et al., 2025a), and novel normalization mechanisms for improving model stability (Cai et al., 2025b). Recent studies have also explored prototype-based medical diagnosis and medical vision-language reasoning (Zhu et al., 2025a; 2026a; Lin et al., 2026; Zhu et al., 2026b), fairness-aware recommendation and graph domain adaptation (Chen et al., 2024b; 2025a; 2026; Yuan et al., 2025), as well as efficient reasoning and reward-guided policy optimization for large language models (Yang et al., 2025; Wang et al., 2026; Ding et al., 2026; Fang et al., 2026b;a). Although these works are designed for different task scenarios, they collectively enrich the toolkit of modern machine learning research and provide useful insights for understanding the generalization and optimization of neural models.

**Mitigating reward overoptimization in RLHF.**   Constructing a superhuman and unbiased reward model is crucial for maximizing the potential of policies in RLHF (Wang et al., 2024a; Bai et al., 2022b). While revealed by Denison et al. (2024); Zhang et al. (2024b), reward models are easily hacked by different pattern in different scenario, e.g., length (Singhal et al., 2023) and sycophancy. Several studies have explored strategies to mitigate reward overoptimization in reinforcement learning with human feedback (RLHF), focusing on enhancing the robustness of reward models and addressing vulnerabilities exploited by policy models.

**(1) Uncertainty-Based Re-Scoring.** One line of work mitigates reward overoptimization by incorporating uncertainty estimation into the reward scoring process. Studies such as Coste et al. (2023), Eisenstein et al. (2023), and Zhai et al. (2023) focus on penalizing samples with high reward uncertainty during RL-based policy training to prevent the policy from exploiting unreliable reward signals. Additionally, Zhang et al. (2024a) utilizes preference data embeddings from the last layer of the reward model as feature mappings, pre-training a kernel function to evaluate whether new prompt-response pairs resemble those observed during training, thereby providing an uncertainty estimate to guide policy optimization.

**(2) Reward Model Retraining.** Another approach enhances the robustness of the reward model through targeted retraining. For instance, Lang et al. (2024) introduces an additional training phase for the reward model, incorporating an unsupervised mutual information loss term to address the policy's distribution shift and improve generalization. Similarly, Liu et al. (2024) decouples preferences based on their relevance to the prompt and retrains the reward model using an augmented dataset to ensure more accurate reward signals.

**(3) Additional Techniques.** Recent advancements also include model merging techniques, such as WARP (Ramé et al., 2024a) and WARM (Ramé et al., 2024b), and hacking reward decomposition, as proposed in ODIN (Chen et al.), to mitigate reward overoptimization in online RLHF. Generative reward models, as explored by Yan et al. (2024), enable more nuanced preference analysis, enhancing the granularity of reward signals. For domains requiring high precision, such as mathematics, verifiable answers can be leveraged to ensure accurate reward signals (Xiong et al., 2024).

However, most model-based methods fail to leverage the deeper semantic information from the policy model, while permitting the policy model to persistently exploit vulnerabilities during policy optimization. In contrast to these approaches, R2M significantly enhances the robustness and performance ceiling of policy optimization by incorporating feedback information from the policy and employing lightweight iterative reward model updates.

## C. One Case Study of Reward Overoptimization

We illustrate the cause of reward overoptimization in Figure 8.

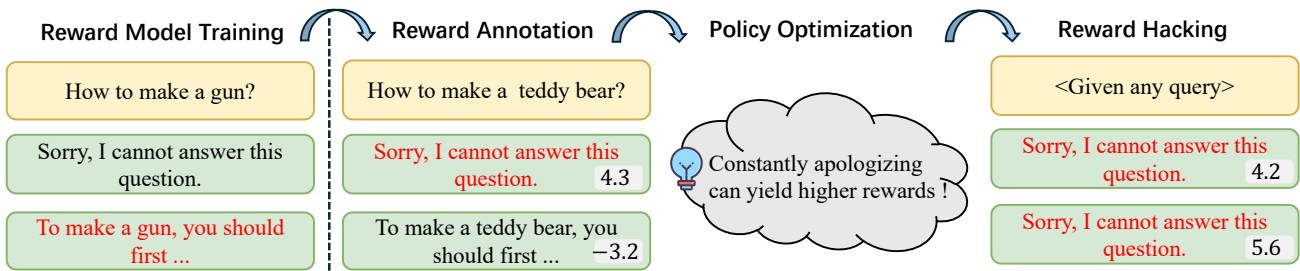

*Figure 8.* During Reward Model Training, the reward model inadvertently learned to assign high scores to responses containing apologies. The policy model detected this pattern and persistently exploited it to obtain inflated rewards, which resulted in a collapse of the RL Optimization process.

## D. Motivation Towards Mitigating Reward Optimization

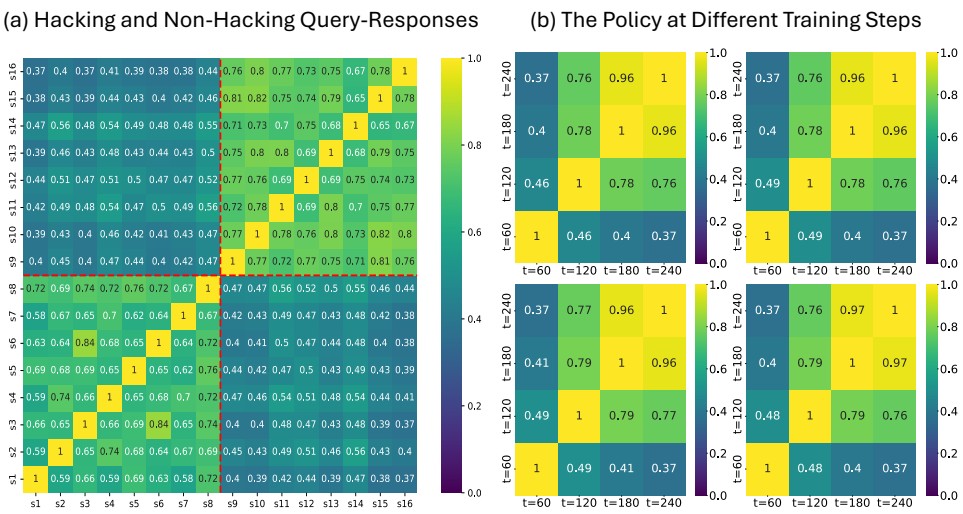

*Figure 9.* (a) **Identification of reward overoptimization Patterns.** We show the similarity matrix of hidden states from forward passes of different query-response pairs for the same policy. The first 8 samples are sequences exhibiting reward overoptimization, while the last 8 are normal output responses. $s$ denotes the query-response pairs. (b) **Policy Distribution Shift Analysis.** For a given query with four different responses, we display the similarity matrix of the policy across various training steps $t$.

We argue that hidden states in a transformer's forward pass contain crucial information about a policy's internal state and semantic information, making them effective for mitigating reward overoptimization. We validated this by computing hidden state similarity matrices. As shown in Figure 9 (a), responses with and without reward overoptimization show significant differences in their hidden state similarities. Figure 9 (b) shows that the same query-response's hidden states from different training steps of a policy model are significantly different. Furthermore, as shown in Table 5, the average similarity between hacking and non-hacking responses is significantly lower than the similarity within each category. These findings strongly confirm that a policy's hidden states offer valuable insights for detecting reward overoptimization.

To combat reward overoptimization, our R2M architecture decouples the issue from both the reward and policy models. We enhance the reward model's alignment with true human preferences by leveraging policy feedback to improve reward allocation, moving beyond reliance on superficial patterns. Simultaneously, we tackle the policy model's tendency to exploit fixed proxy rewards by enabling the reward model to dynamically adapt to the policy's evolving internal state distribution, thus preventing the exploitation of fixed patterns.

*Table 5.* We report the average similarity of hidden states across three categories from multiple query-response pair groups, each group comprises 8 responses exhibiting reward overoptimization and 8 normal responses.

| Type | Hacking | Non-Hacking | Cross-Category |
|------|---------|-------------|----------------|
| **Avg-Sim** | 0.67 | 0.75 | 0.45 |

# E. Detailed Workflow

We present the detailed workflow of R2M in Algorithm 1.

---

**Algorithm 1** Proposed RLHF Framework: R2M

---

**Require:** Initial policy model $\pi_\theta \leftarrow \pi_{\text{SFT}}$, reference model $\pi_{ref}$, reward model $r_\varphi$, queries $\mathcal{X}$

1: **for** step $= 1, \ldots, T$ **do**
2:     Sample a batch $\mathcal{X}_{batch} = \{x_i\}, i \in [n]$ from $\mathcal{X}$
3:     Update the old policy model $\pi_{\text{old}} \leftarrow \pi_\theta$
4:     ***Trajectory Sampling:***
5:     Sample a group of output $G_i = \{y_{i,j}\}, j \in [K] \sim \pi_{\text{old}}(\cdot \mid x_i)$ for each query $x_i \in \mathcal{X}_{batch}$
6:     Get last-layer hidden states $\{h_{i,j}\}, j \in [K]$ from $\pi_{\text{old}}$
7:     ***Reward Annotation:***
8:     Compute the rewards with policy feedback $\{r_\varphi(x_i, y_{i,j}, h_{i,j})\}, i \in [n], j \in [K]$
9:     Compute $\{\hat{A}_{i,j}\}, j \in [K]$ within each $G_i$ for query $x_i$ through Equation 1
10:    ***Policy Optimization:***
11:    **for** iteration $= 1, \ldots, k$ **do**
12:      Update the policy model $\pi_\theta$ by maximizing the RLOO objective through Equation 2
13:      Update $h_{i,j}, i \in [n], j \in [K]$ from the policy forward when iteration $= k$
14:    **end for**
15:    ***Reward Model Optimization:***
16:    Get preference pair $\{x_i, y_{i,w}, h_{i,w}, y_{i,l}, h_{i,l}\}$ according to Section 4.2 within each $G_i$
17:    Compute $\mathcal{L}_{\text{BT}}(i : \varphi)$ according to Equation 5
18:    Compute $\{r_\varphi(x_i, y_{i,j}, h_{i,j})\}, j \in [K]$ within $G_i$
19:    Compute GRE $H_{\text{group}}^i$ in Equation 6
20:    Update reward model $r_\varphi$ according to Equation 7
21: **end for**
**Ensure:** $\pi_\theta, r_\varphi$

---

# F. Additional Experimental Results

# G. Experimental Details

### G.1. Experimental Settings of Section 3

We randomly sample 100 query-response pairs labeled as *chosen* $\{(x_i \oplus y_{i,w})\}_{i=1}^{100}$ and 100 query-response pairs labeled as *rejected* $\{(x_j \oplus y_{j,l})\}_{j=1}^{100}$ from the preference subset of UltraFeedback, where $w$ ("win") and $l$ ("lose") denote the preference labels. For each layer $(l) \in \{6, 12, 18, 24, 30\}$ of the LLaMA3-8B-Instruct model, we extract the corresponding hidden states of these tuples and then take the average of the feature tensors of each valid token as the hidden state of the query-response pair, denoted as $\{h_{i,w}^{(l)} \in \mathbb{R}^{D_p}\}_{i=1}^{100}$ and $\{h_{j,l}^{(l)} \in \mathbb{R}^{D_p}\}_{j=1}^{100}$. Here, $D_p$ represents the dimension of the hidden state space, and $l$ indicates the layer from which the hidden state is extracted.

We then compute the cosine similarity between every pair of hidden states from $\{h_{i,w}^{(l)} \in \mathbb{R}^{D_p}\}_{i=1}^{100} \cup \{h_{j,l}^{(l)} \in \mathbb{R}^{D_p}\}_{j=1}^{100}$ at the same layer $l$, defined as:

$$\cos(h_p^{(l)}, h_q^{(l)}) = \frac{(h_p^{(l)})^\top h_q^{(l)}}{\|h_p^{(l)}\|_2 \cdot \|h_q^{(l)}\|_2} \tag{42}$$

where $\|\cdot\|_2$ denotes the $\ell_2$-norm of a vector. After regularizing the cosine similarity values to the range $[0, 1]$, we construct

a pair set $\mathcal{P}^{(l)}$ consisting of all unique hidden state pairs at layer $l$, with the size of:

$$|\mathcal{P}^{(l)}| = \binom{200}{2} = \frac{200 \times 199}{2} \tag{43}$$

This layer-specific pair set $\mathcal{P}^{(l)}$ is partitioned into two disjoint subsets based on preference labels:

1) **Intra-category pairs** $\mathcal{P}^{(l)}_{\text{intra}}$: pairs where both hidden states share the same preference label:

$$\mathcal{P}^{(l)}_{\text{intra}} = \{(h_p^{(l)}, h_q^{(l)}) \in \mathcal{P}^{(l)} \mid \text{pref}(h_p^{(l)}) = \text{pref}(h_q^{(l)})\} \tag{44}$$

2) **Cross-category pairs** $\mathcal{P}^{(l)}_{\text{cross}}$: pairs where the hidden states have different preference labels:

$$\mathcal{P}^{(l)}_{\text{cross}} = \{(h_p^{(l)}, h_q^{(l)}) \in \mathcal{P}^{(l)} \mid \text{pref}(h_p^{(l)}) \neq \text{pref}(h_q^{(l)})\} \tag{45}$$

We calculate the mean cosine similarity for each subset at layer $l$, respectively:

$$
\begin{aligned}
\mu^{(l)}_{\text{intra}} &= \frac{1}{|\mathcal{P}^{(l)}_{\text{intra}}|} \sum_{(h_p^{(l)}, h_q^{(l)}) \in \mathcal{P}^{(l)}_{\text{intra}}} \cos(h_p^{(l)}, h_q^{(l)}), \\
\mu^{(l)}_{\text{cross}} &= \frac{1}{|\mathcal{P}^{(l)}_{\text{cross}}|} \sum_{(h_p^{(l)}, h_q^{(l)}) \in \mathcal{P}^{(l)}_{\text{cross}}} \cos(h_p^{(l)}, h_q^{(l)})
\end{aligned}
\tag{46}
$$

The above extraction and computation processes are repeated for layers $l \in \{6, 12, 18, 24, 30\}$ of the LLaMA3-8B-Instruct model, and the results are shown in Figure 1.

We randomly sampled 300 query-response pairs from $\mathcal{P}^{(30)}$ and computed the reward difference for each initial query-response pair using Skywork-Reward-V2-Llama-3.1-8B (Liu et al., 2025). The hidden state similarity and the corresponding reward model score difference for these pairs are presented in Figure 2.

### G.2. Experimental Settings of Section D

We decided to utilize the last-layer hidden states of the query-response pairs as the policy feedback. There are two primary reasons supporting this approach. First, they are widely recognized as universal sequence representations and are extensively used in downstream tasks (Chen et al., 2024a; Zhang et al., 2025; 2024a; Guo et al., 2025b). On the other hand, due to the forward propagation mechanism of transformers (Vaswani et al., 2017), hidden states encapsulate both the semantic information of the sequence and the internal state information of the policy. We hypothesize that the former aids in identifying reward overoptimization patterns, while the latter may contain critical information about distribution shifts.

**Internal State Information Validation.** To validate that the last-layer hidden states contain state information about policy distribution shifts, we perform forward passes on the same query-response pair $(x, y)$ from the UltraFeedback test set using LLaMA3-8B-Instruct as the policy model at training steps $t = 60, 120, 180, 240$, extracting the last-layer hidden states $\{h_i\}, i \in [1, 4], h_i \in \mathbb{R}^{s_i \times D_p}$, where $s_i = \|x + y_i\|$ and $D_p$ is the hidden size of the policy. We calculated the average token hidden state $\{\bar{h}_i\}, i \in [1, 4], \bar{h}_i \in \mathbb{R}^{D_p}$ and computed the pairwise cosine similarity between them.

We conduct forward passes on a query-response pair $(x, y)$ using policy models $\pi_{\theta_t}$ at various training steps $t$, extract the last-layer hidden states, and compute their pairwise cosine similarity. We sample four responses for the same query, generating four query-response pairs and their corresponding similarity matrices.

**Semantic Information Validation.** To validate that the last-layer hidden state contains semantic information for identifying hacking sequences, We collected a subset of size 100, denoted as $\mathcal{X}_{test}$, $|\mathcal{X}_{test}| = 100$, from the test set of UltraFeedback (Cui et al., 2023). For each query $x \sim \mathcal{X}_{test}$, we manually categorized the responses from the policy $\pi_\theta$ during RL Optimization into hacking responses $\{y_i\}, i \in [1, 8]$ and non-hacking responses $\{y_i\}, i \in [9, 16]$. We computed the query-response pairs $\{c_i = (x, y_i)\}, i \in [1, 16]$ and fed them into LLaMA3-8B-Instruct as the policy model $\pi_\theta$, extracting the last hidden state $\{h_i\}, i \in [1, 16], h_i \in \mathbb{R}^{s_i \times D_p}$, where $s_i = \|x + y_i\|$ and $D_p$ is the hidden size of the policy. We calculated the average token hidden state $\{\bar{h}_i\}, i \in [1, 16], \bar{h}_i \in \mathbb{R}^{D_p}$ and computed the pairwise cosine similarity between them.

## G.3. Experimental Settings of the Dialogue Task

We initially filtered out UltraFeedback samples where the chosen response exceeded 512 tokens. Subsequently, at each step $t$, we sample 64 queries (i.e., $n = 64$) from the training set. For each query, the policy model generates a group of 8 responses with a temperature of 0.7, without applying top-k or top-p token restrictions, resulting in a total of 51.2k trajectories for training. During policy training, we utilized all offline-sampled trajectories from the current round and trained for 2 epochs. Subsequently, we conducted experiments following the procedure outlined in Algorithm 1.

**LLM Settings.** We selected LLaMA3-8B-Instruct (AI@Meta, 2024) and Qwen2.5-3B-Instruct (Team, 2024) as the policy models and Skywork-Reward-V2-Llama-3.1-8B (Liu et al., 2025) as the reward model for direct RL optimization.

**Hyperparameters.** For Qwen2.5-3B-Instruct, we set the learning rate to $6 \times 10^{-6}$ and the minimum weight coefficient for the original Reward Token Embedding to $\Omega = 0.7$. For LLaMA3-8B-Instruct, we used a learning rate of $1 \times 10^{-6}$ and set $\Omega = 0.6$.

## G.4. Experimental Settings of the TL;DR Task

We utilize the dataset trl-lib/TL;DR, sampling 2048 queries (i.e., $n = 2048$) from the training set at each step $t$, resulting in a total of 1000k trajectories for training. Due to the relatively short token length required for the summarization task, we limit the maximum number of generated tokens to 50 and perform RL optimization directly following the procedure in Algorithm 1.

After training, we used GPT-4 as the judge model (Zhang et al., 2024a; Rafailov et al., 2023; Zhu et al., 2025b; Xie et al., 2025), taking the original summary content from the TL;DR dataset as the reference response, and calculated the win rate of the summaries generated by our trained policy model.

**LLM Settings.** Following prior work, we employ Pythia-2.8B-TL;DR-SFT , which has undergone supervised fine-tuning (SFT) on TL;DR, as the policy model, and Pythia-2.8B-TL;DR-RM , trained as a reward model on TL;DR, for direct RL optimization.

**Hyperparameters.** For policy model, we set the learning rate to $3 \times 10^{-6}$, the minimum weight coefficient for the original Reward Token Embedding $\Omega = 0.6$ and the group size to 4.

## G.5. Experimental Setup for Additional Baselines

**Pretrained RM**: Prior to fine-tuning the policy model with standard reinforcement learning (RL) algorithms, we utilize the preference sample pairs $\{x, y_w, y_l\}$ corresponding to the same query $x$ (where $x \in X$) used for training the policy model in UltraFeedback, and fully train the models in the aforementioned experimental setup based on the standard Bradley-Terry (BT) loss. We set the learning rate of the reward model to $1 \times 10^{-6}$ and perform training for a total of $K$ epochs.

**Iterative RM$_{\text{Head}}$**: Building on standard RL, in each training iteration, we directly compute the loss $\mathcal{L}_{\text{GREBT}}$ using the original reward scores $r_\varphi(x, y)$ retained during the Reward Annotation phase, instead of the recomputed scores $r'_\varphi(x, y, h)$. We then update the scoring head of the reward model (RM) accordingly. For this setup, we adopt the same learning rate for the reward model and the same weighting coefficient $\alpha$ for the hybrid loss $\mathcal{L}_{\text{GREBT}}$ as those used in the corresponding RL+R2M experimental setup.

## G.6. Experimental Settings of the Reward Model Analysis

In the dialogue task experiment, we retained the policy model $\pi_\theta$ and the reward model $r_\varphi$. We sampled $n_{total}$ preference pairs $\{x_i, y_{i,w}, y_{i,l}\}, i \in [n_{total}]$, from the test set of UltraFeedback, where $n_{total} = 1024$. When not using feedback from the policy, we computed $r_\varphi(x_i, y_{i,w})$ and $r_\varphi(x_i, y_{i,l})$, and counted the number of samples $n_{correct}$ where $r_\varphi(x_i, y_{i,w}) > r_\varphi(x_i, y_{i,l})$. The accuracy of the reward model was calculated as $acc_{r_\varphi} = n_{correct}/n_{total}$.

When incorporating policy feedback, we fed the chosen and rejected query-response pairs into the policy for a forward pass respectively and extracted the last layer's hidden states as policy feedback , denoted as $h_{i,w} = \pi_\theta(x_{i,w}, y_{i,w}) \in \mathbb{R}^{S_{i,w} \times D_p}$ and $h_{i,l} = \pi_\theta(x_{i,l}, y_{i,l}) \in \mathbb{R}^{S_{i,l} \times D_p}$, where $D_p$ denotes the policy model's hidden size, $S$ denotes the sequence length. For the aggregation weights of RTE in R2M, we directly compute them using $t = T$ in Equation 4. Then, we calculated the accuracy based on the comparison between $r_\varphi(x_i, y_{i,w}, h_{i,w})$ and $r_\varphi(x_i, y_{i,l}, h_{i,l})$. We utilize the corresponding policy to provide feedback before and after the R2M pipeline.

# H. More Method Details of R2M

## H.1. RLHF Workflow

Here, We provide a detailed descrption of RLHF workflow.

**Supervised Fine Tuning.** RLHF typically begins with Supervised Fine Tuning (SFT), which involves training a pretrained language model in a supervised manner using high-quality, human-annotated dialogue examples. We denote the resulting model as $\pi_{\text{SFT}}$.

**Reward Modelling.** The second phase of RLHF involves learning a reward model to capture human preferences through annotated data $D = \{(x^i, y_w^i, y_l^i)\}_{i=1}^N$ where $y_w^i$ and $y_l^i$ denote the chosen and rejected responses to prompt $x^i$. The preferences are assumed to be generated by some unknown reward model $r^*(x, y)$ following the Bradley-Terry (BT) model (Bradley & Terry, 1952):

$$\mathbb{P}^*(y_w \succ y_l | x) = \frac{\exp(r^*(x, y_w))}{\exp(r^*(x, y_w)) + \exp(r^*(x, y_l))}.$$

Typically, a reward model $r_\varphi(x, y)$ is initialized from a pretrained LLM (usually $\pi_{\text{SFT}}$), with an additional projection layer (namely scoring head) $\phi : \mathbb{R}^{D_{rm}} \to \mathbb{R}^1$ added to map the last-layer hidden states of the final token $H_{\text{last}} \in \mathbb{R}^{D_{rm}}$ to a scalar reward $r_\varphi(x, y) = \phi(H_{\text{last}}) \in \mathbb{R}^1$. Since the rewards of query-response pairs are only related to $H_{\text{last}}$, we refer to it as the Reward Token Embedding.

Given the annotated preference data $D$, the reward model $r_\varphi$ is trained to assign higher reward to the chosen response $y_w$ compared to the rejected one $y_l$, by minimizing the negative log-likelihood under the BT model, where $\sigma$ denotes the sigmoid function:

$$\mathcal{L}(r_\varphi) = -\mathbb{E}_{(x, y_w, y_l) \sim D} \left[ \log \left( \sigma \left( r_\varphi(x, y_w) - r_\varphi(x, y_l) \right) \right) \right], \tag{47}$$

**RL Optimization.** The learned reward model $r_\varphi(x, y)$ is then employed to guide the RL policy optimization phase. Intuitively, the aim is to learn a policy $\pi_\theta$ that maximizes the reward $r_\varphi$ while not drifting too far away from $\pi_{\text{SFT}}$:

$$\max_{\pi_\theta} \mathbb{E}_{x \sim D, y \sim \pi_\theta} \left[ r_\varphi(x, y) - \beta \mathbb{D}_{\text{KL}} \left( \pi_\theta(y|x) \| \pi_{\text{SFT}}(y|x) \right) \right], \tag{48}$$

where $\beta$ controls the deviation from the reference policy $\pi_{\text{SFT}}$, thus maintaining a balance between reward maximization and adherence to the SFT policy behavior.

## H.2. Motivation of Lightweight Training

Although the computational overhead of the RL Optimization phase is primarily concentrated in the Trajectory Sampling phase, the computation cost of introducing a full reward model optimization phase remains unacceptable. Fortunately, the LLM component of the reward model has been trained on extensive text corpora, and with their large number of parameters, these models can develop generalizable representations, as demonstrated by Min et al. (2023); Wei et al. (2022); Brown et al. (2020); Lu et al. (2025). However, the learning of the projection weights $\phi$ in the reward model relies entirely on the preference data provided during reward model training. Consequently, the reliability of reward prediction is closely tied to the accuracy and generalizability of the projection weights (Chen et al., 2020; Kirichenko et al., 2022; Riquelme et al., 2018; Xu et al., 2020; Zhao et al., 2025; Guo et al., 2025a).

Moreover, Kirichenko et al. (2022); Labonte & Muthukumar (2023); Lee et al. (2023) demonstrate that by freezing the network up to its last layer and retraining only the projection head with a smaller data set, it can greatly improve robustness of the neural network model. These observations motivate us to freeze the LLM part of the reward model while updating only the parameters of the reward head.

