# OpenReview forum: "Real-Time Aligned Reward Model beyond Semantics"
_ICML.cc/2026/Conference — ICML 2026 regular_

### Official Review · Reviewer_B1hX · 2026-02-24

**Soundness:** 3
**Presentation:** 3
**Significance:** 3
**Originality:** 3
**Overall Recommendation:** 4
**Confidence:** 3

**Summary:**

This paper introduces R2M, a lightweight RLHF framework designed to mitigate reward overoptimization by incorporating the policy model's evolving hidden states into the reward model (RM). By freezing the RM backbone and iteratively updating the scoring head using a novel Group Reward Entropy Bradley-Terry (GREBT) loss, the method achieves real-time alignment with the policy's distribution shift, demonstrating significant improvements on AlpacaEval 2 and TL;DR benchmarks compared to standard RLOO and GRPO baselines.

**Compliance With Llm Reviewing Policy:**

Affirmed.

**Final Justification:**

The authors have provided convincing counter-arguments and excellent practical solutions to the limitations I raised. The method is novel, lightweight, and the empirical gains are solid. I am maintaining my score of Weak Accept (and leaning towards a solid Accept). I believe this paper will be a valuable and practical contribution to the RLHF alignment community.

**Key Questions For Authors:**

1. If the policy model begins to exploit spurious patterns (reward hacking), feeding its internal "hacking" states back into the reward model might inadvertently reinforce these behaviors rather than correct them.

2. The Group Reward Entropy (GRE) loss forces score differentiation within a group; however, in late-stage training where responses are uniformly high-quality, this may introduce artificial noise and penalize convergence.

3.The requirement to transfer large hidden state tensors from the policy to the reward model during generation creates significant communication bottlenecks in typical distributed RLHF infrastructures, which is not fully addressed in the cost analysis.

**Limitations:**

The authors have included a "Broader Impact" section, but they have **not** adequately discussed the technical **limitations** of their proposed method.

Suggestions for improvement:

1.The paper claims the method is lightweight based on GPU computation, but it overlooks the **communication cost**. In standard distributed RLHF frameworks (e.g., using Ray or OpenRLHF), the Policy and Reward Models often reside on different nodes. Transmitting high-dimensional hidden state tensors (Batch × Sequence Length × Hidden Dimension) is significantly more bandwidth-intensive than transmitting token IDs. The authors should discuss this engineering bottleneck.

2.The method relies on the assumption that the policy's hidden states encode useful "implicit reward" information (aligned with DPO findings). The authors should discuss potential limitations when the policy model is significantly smaller or weaker than the reward model, or if the policy has not yet learned meaningful representations of the task.

3.While the goal is to mitigate reward hacking, there is a theoretical risk that the Reward Model might learn to associate specific "hacking" patterns in the Policy's hidden states with high scores, thereby reinforcing the hacking behavior rather than correcting it. A discussion on the robustness of the alignment in such adversarial scenarios would be beneficial.

**Strengths And Weaknesses:**

1.The proposal to leverage the policy's deep hidden states as "policy feedback" to augment the reward model is a fresh and theoretically motivated perspective, effectively bridging implicit and explicit reward modeling.

2.The framework provides a practical solution for iterative reward modeling by updating only the lightweight scoring head and cross-attention modules, avoiding the prohibitive cost of full RM retraining.

3.The method demonstrates consistent and significant gains over strong baselines (RLOO, GRPO) across multiple metrics, including a 5.2% - 8.0% increase in AlpacaEval 2 win rates.

---

> ### Author Rebuttal · Authors · 2026-03-31
>
> Thank you for your valuable feedback. We anwsered each of your detailed questions below. We hope this resolves your concerns, and we are happy to respond any further questions you may have.
>
> ---
>
> > **Q1 & L3:** policy feedback may degrade reward model performance
>
> Thank you for this highly insightful question. However, R2M’s mechanism design already limits such adversarial risks through the following safeguards.
>
> **The frozen LLM component retains the core capabilities of RM:**
> During reward-model updates we freeze the LLM component (Appendix H.2), fully preserving the general representational capabilities acquired during the reward model’s initial pre-training phase. This ensures that even if the policy model generates specific “tampering” patterns in its hidden states, updates to the cross-attention module and scoring head remain strictly anchored to the Reward Token Embedding that conforms to human preferences (Appendix H.1, Lines 1218–1221). Consequently, the RM retains its fundamental semantic discrimination ability against superficial cheating behaviors (e.g., repeatedly apologizing or stacking specific vocabulary).
>
>
>
> ---
>
> > **Q2:** concern relative to the GRE loss
>
> **The GRE loss activates only when genuine quality differences exist.** It sharpens reward distributions by amplifying existing score gaps without altering the correct ranking, thus alleviating group degeneration rather than creating artificial distinctions. When response qualities are identical, **the GRE loss gradient is exactly zero**: it cannot artificially inflate score differences (reviewer **ZV5Q, Q4**).
>
> ---
>
> > **Q3 & L1**: multi-node communication bottleneck
>
> We acknowledge that cross-node communication is a valid concern for distributed deployment. However, this overhead can be substantially reduced by design.
>
> **Reducing communication volume.** Rather than transmitting the full hidden state tensor ($B \times S \times D_p$), we can place a lightweight copy of the cross-attention module on the policy node (parameter count: only $2 \times (D_r \times d + D_p \times d)$). During rollout, only the RM's reward token embedding ($B \times D_p$) is sent to the policy node for cross-attention aggregation, and the updated embedding is sent back. This reduces per-step communication from $B \times S \times D_p$ to $2 \times B \times D_p$ — critically, **the cost no longer scales with sequence length $S$**.
>
> **Asynchronous updates.** Since the RM update is faster than policy optimization, the updated cross-attention module can be synchronized to the policy node while the policy continues training, avoiding network blocking.
>
> With these optimizations, R2M adds only $2 \times B \times D_p$ in data transfer during rollout, and training-time communication remains identical to standard RL. We will add this discussion to the paper.
>
> ---
>
> > **L2:** the effectiveness of policy feedback when the policy model is weak.
>
> We fully understand the reviewer’s concern that policy hidden-state quality may affect R2M effectiveness. In practice, LLM-based RL typically requires the base model to have strong prior capability, so the policy is usually initialized from an SFT checkpoint. This ensures the policy’s hidden states carry sufficient semantic information from the start and can effectively support reward-model judgments.
>
> ---
> > **Q4**: Semantic mapping from policy to RM can be difficult to learn
>
> We want to highlight that the cross-space fusion gap is actually limited and RL only leads to relatively sparse hidden-state changes.
>
> **The semantic gap between RMs and policy models is narrow:**
> The deep-layer hidden states of the policy model implicitly encode preference relations (Section 3). The similarity of policy hidden states for response pairs strongly correlates with reward score differences (Figure 2), indicating substantial similarity between their semantic spaces. Since the RM uses a pure LLM backbone and prior work regularizes hidden states via SFT loss mixing during RM training [1], the cross-space fusion gap is actually limited.
>
> **Hidden state updates of the policy model are sparse in the RL process:**
> [2] demonstrates that, compared with the substantial representational shift induced by SFT, RL achieves retention of pre-training capabilities while activating reasoning ability through sparse adjustments to hidden-state representations.
>
> These points provide strong justification that learning the semantic-space mapping from policy to RM remains feasible even when only the small-parameter cross-attention module and scoring head are trainable.
>
>
>
>
>
> References:
>
> [1] https://arxiv.org/abs/2507.07981
>
> [2] https://arxiv.org/abs/2509.04259

---

> > ### Author Rebuttal · Reviewer_B1hX · 2026-04-01
> >
> > The authors have provided convincing counter-arguments and excellent practical solutions to the limitations I raised. The method is novel, lightweight, and the empirical gains are solid. I am maintaining my score of **Weak Accept** (and leaning towards a solid Accept). I believe this paper will be a valuable and practical contribution to the RLHF alignment community.

---

> > > ### Author Response · Authors · 2026-04-01
> > >
> > > Dear Reviewer B1hX,
> > >
> > > Thank you very much for acknowledging that our responses have fully resolved your concerns, and for your positive assessment of R2M's contribution to the RLHF alignment community. Your detailed and constructive feedback has been invaluable in strengthening our work. We will definitely incorporate all of your suggestions and discussions into the revised manuscript.
> > >
> > > Given this, we would be very grateful if you could consider adjusting your score to reflect your stated inclination of "leaning towards a solid Accept." Your support would be very meaningful for this work.
> > > Thank you again for your time and thoughtful review!
> > >
> > >
> > > Best regards,
> > >
> > > Authors

---

### Official Review · Reviewer_jQVC · 2026-03-03

**Soundness:** 2
**Presentation:** 3
**Significance:** 2
**Originality:** 3
**Overall Recommendation:** 4
**Confidence:** 5

**Summary:**

The paper attempte to address the reward overoptimization in RLHF, which is primarily driven by the continuous distribution shift of policy against a static, frozen reward model. To tackle this, the authors proposed Real-Time Aligned Reward Model (R2M), a lightweight framework designed to iteratively update the RM during RL optimization.

**Compliance With Llm Reviewing Policy:**

Affirmed.

**Final Justification:**

The reviewers solved all my concerns, so I would like to raise my score.

**Key Questions For Authors:**

- Although it is a sensible way to fuse heterogeneous information using cross attention, projecting policy hidden states directly into a RM's space might be challenging to learn, especially when the policy output distribution change largely.
- The design of GRE loss, minize entropy on rewards could also artificially inflate score differences even when responses are genuinely identical in quality, potentially leading to instability or a new form of hacking.
- For the reward curve in Fig 5 and Fig 6, whether the reward is computed by a third-party reward, or corresponding different RMs used for training?
- Is it meaningful to compare the loss in Fig 5? The policy loss is directly computed using the estimated advantages, which are derived from the rewards. So the lower loss does not definitively prove better optimization, it could simply be a mathematical artifact of how R2M's RM scales its outputs or how GRE loss affect the variance of the rewards within a group.
- See weakness above.

**Limitations:**

- Insufficient empirical analysis. The current analysis is limited by short training horizons.
- Practical deployment challenges. There remain a major gap in applying this framework to practical, distributed training scenarious.

**Strengths And Weaknesses:**

Strengths:
- The framework of R2M, injecting policy's hidden states to RM and update RM during RL, is novel.
- The proposed GREBT loss is an interesting addition that force the RM to maintain score disparities.
- The empirical validation demonstrating the effective of R2M compared with baseline.

Weakness:
- Lack of stability analysis. Optimizing two models at the same time, intuitively, is less stable than optimize single model. All experiments are conducted only 100 steps, and the reward curve not seems to convergence for 7B model, making it unknow of the stability of R2M for further training.
- The benchmark results not fully support authors claim. The results on MT-Bench seems not working for RL baseline, not sure if the baseline experiments are conduct correctly, also more benchmark is required.
- Limited scope of application. The R2M framework appears to design for preference optimization scenarios. It is not clear how this method apply to reasoning tasks such as mathematical or code, which is the main focus of current research.
- Scalability and Distributied of R2M. The experiments are limied in relative small models (Qwen2.5-3B and LLaMA3-8B), and the cost analysis is conducted under single GPU. R2M requires capturing the last-layer hidden states from the policy, it will become extremely large with the increase of model size and sequence length, if the policy and reward models are hosted on seperate nodes/GPUs (which is standard for large LLMs training), in this case, the network bandwidth will become a bottleneck, rather than FLOPs. Making it unclear if this is a scalability framework and the calim of "negligible overhead" might fall apart in a realistic.

---

> ### Author Rebuttal · Authors · 2026-03-31
>
> Thank you very much for your valuable feedback. Due to space limitations, we have moved part of the response to other reviewers' section. Please refer there for details, and we look forward to your further feedback.
>
> ---
> > **W1 & L1:**  training stability and convergence
>
> **Stability:**
> Thank you for raising this concern. We address the instability from two angles:
>
> First, R2M and the policy are trained sequentially (not simultaneously)—the policy is updated first, then R2M is updated. This decoupled schedule avoids the instability typically associated with joint optimization.
>
> Second,  R2M is designed to enhance, not undermine, the training stability. The primary source of instability in RL training is reward signals that diverge from true human preferences as the policy distribution shifts. R2M directly addresses this by realigning the reward model in real time.
>
> **Convergence**: Despite limited training steps, our large batch size (512) ensures sufficient question exploration and convergence. For LLaMA3 (Figure 5), we further trained 100 steps and evaluated Alpaca-Eval every 20 steps to verify convergence.
>
> | Step | RLOO | RLOO+R2M |
> |------|------|----------|
> | 100  | 28.4 | 34.5     |
> | 120  | 28.3 | 34.2     |
> | 140  | 28.1 | 34.4     |
> | 160  | 28.2 | 34.6     |
> | 180  | 28.4 | 34.2     |
> | 200  | 28.2 | 34.4     |
>
> Results show that Alpaca-Eval performance across all checkpoints varies only slightly, demonstrating that the experiments had converged by step 100.
>
> ---
>
> > **W2:** the reasonableness of evaluation benchmarks
>
> We emphasize that GRPO and RLOO are equally applicable to RLHF and RLVR. The only difference is that our reward is a scalar score given by a LLM-based RM rather than rule-based reward.
>
> MT-Bench is a general benchmark for RLHF, with scores dependent only on models rather than training methods. R2M is trained on UltraFeedback and evaluated on MT-Bench and Alpaca-Eval, which is a widely adopted setup ([1], [2], [3]). We thus consider MT-Bench a reasonable evaluation choice.
>
> In addition, we have incorporated the Arena-Hard results (Qwen2.5-72B-Instruct as the judge model):
>
> | Model                | SFT  | RLOO | RLOO+R2M | GRPO | GRPO+R2M |
> |----------------------|------|------|----------|------|----------|
> | Qwen2.5-3B-Instruct  | 6.8  | 10.7 | 14.7     | 11.2 | **15.8** |
> | LLaMA3-8B-Instruct   | 9.9  | 16.1 | 19.5     | 16.8 | **20.3** |
>
> After deploying R2M, the policy obtains consistent and significant gains on Arena-Hard, further confirming the effectiveness of our method.
>
> ---
>
> > **W3:** extension to math and code tasks
>
> Math has unique correct answers, while code can be evaluated via interpreters. In both scenarios (RLVR), rewards are directly obtainable through rule-based verifiers, making LLM-based RM unnecessary.
>
> R2M also has broad applicability in RLVR. Current RLVR lacks process supervision for long CoTs. The process reward model (PRM), which assigns step-level scores, shares the same base architecture as R2M, they are fully compatible. We plan to extend R2M to PRMs in future work to deliver more accurate process supervision for long CoTs.
>
> ---
>
> > **W4 & L2:** multi-node communication cost
>
> This issue is resolved in Response to reviewer **B1hX Q3 & L1** (Please refer there for details). After optimization, R2M only adds $2 \times B \times D_p$ data transfer during rollout, with training time identical to standard RL.
>
> ---
>
> > **Q1:** Semantic mapping from policy to RM can be difficult to learn
>
> We want to highlight that the cross-space fusion gap is actually limited  and RL only leads to relatively sparse hidden-state changes (Please refer Reviewer **B1hX, Q4** for details). These points provide strong justification that learning the semantic-space mapping from policy to RM remains feasible even when only the small-parameter cross-attention module and scoring head are trainable.
>
>
> ---
> > **Q2:** GRE loss might affect scoring fairness
>
> **The GRE loss activates only when genuine quality differences exist.** It sharpens reward distributions by amplifying existing score gaps without altering the correct ranking, thus alleviating group degeneration rather than creating artificial distinctions. When response qualities are identical, **the GRE loss gradient is exactly zero**: it cannot artificially inflate score differences (reviewer **ZV5Q, Q4**).
>
> ---
>
> > **Q3:** meaning of reward curves
>
> Both Figure 5 and Figure 6 use the RM that was active during training.
>
> ---
>
> > **Q4:** meaning of losses in Figure 5
>
> We fully agree that policy loss alone cannot directly prove superior alignment. We therefore **do not** argue R2M's advantages based on loss values. The loss curves in Figure 5 only complement reward and KL divergence curves to illustrate full training dynamics.
>
> ---
> References:
>
> [1] https://arxiv.org/abs/2409.13156
>
> [2] https://arxiv.org/abs/2410.10148
>
> [3] https://arxiv.org/abs/2405.14734

---

> > ### Author Rebuttal · Reviewer_jQVC · 2026-04-01
> >
> > Some of my concerns have been addressed by the authors; however, several questions still remain.
> >
> > 1) I still have difficulty understanding the statement in the response to W2 that the scores are “dependent only on the model rather than the training method.” This point needs further clarification. More importantly, my main concern remains: why do baseline methods such as GRPO and REINFORCE++ underperform the SFT model?
> >
> > 2) If Fig.5 and Fig.6 use the reward model active during each method’s training, then RLOO and R2M are being evaluated under different reward functions. In that case, absolute reward and policy-loss comparisons are not directly meaningful. I would need evaluation under a common frozen third-party evaluator or judge.
> >
> > Therefore, I will maintain my current score.
> >
> > Edit: I thank for the clarifications from authors, all my questions are solved, and I would raise my score to weak accept. But I still consider the display of Fig.5/6 and corresponding statements in the main paper are very confusing, and suggest to adjust these parts.

---

> > > ### Author Response · Authors · 2026-04-01
> > >
> > > We appreciate the reviewer's continued engagement on this point. We would like to provide further clarification.
> > >
> > > > Q1:  Why do baseline methods such as GRPO and REINFORCE++ underperform the SFT model?
> > >
> > > Regarding the statement that "dependent only on the model rather than the training method": what we intended to convey is that MT-Bench scores reflect the overall capability of the resulting model checkpoint, rather than being a direct measure of the effectiveness of a particular training algorithm.
> > >
> > > Regarding why baseline methods such as GRPO and REINFORCE++ occasionally underperform the SFT model on MT-Bench: this phenomenon is not unique to our work and has been documented in prior literature. For instance, SimPO ([Meng et al., 2024](https://arxiv.org/pdf/2405.14734)) reports a similar observation where RL-tuned models underperform their SFT counterparts on certain benchmarks. The underlying cause is that preference optimization with a reward model can introduce distributional biases — the policy may overfit to patterns favored by the reward model (e.g., verbosity, hedging) that do not align with MT-Bench's GPT-4-based evaluation criteria. This is, in fact, a manifestation of the reward model misalignment problem that our work aims to address.
> > >
> > > Furthermore, Alpaca-Eval is a primary evaluation benchmark as well.
> > > As shown in Table 1, although REINFORCE++ and ReMax achieve marginally lower MT-Bench scores than the SFT model, they significantly outperform the SFT model on Alpaca-Eval.
> > > This confirms that our baselines are properly implemented and effective. Notably, R2M further improves MT-Bench scores over all baselines, suggesting that the proposed method helps mitigate the very distributional biases that cause baseline degradation on this benchmark.
> > >
> > > > Q2: A common frozen third-party evaluator or judge.
> > >
> > >
> > > We fully agree that since R2M modifies the reward model during training, the reward curves in Figures 5 and 6 reflect training dynamics under different reward functions and cannot be used to directly compare final policy quality between RLOO and RLOO+R2M. We want to clarify that this was indeed our original intention. We never used absolute reward values as a proxy for policy performance. Specifically, Figure 5 illustrates that "R2M maintains reward consistency while allocating higher rewards," and Figure 6 shows that "R2M effectively encourages more aggressive policy updates by assigning systematically higher rewards." These descriptions characterize the training dynamics, not the final policy quality.
> > >
> > >
> > >
> > > To directly address the reviewer's concern, we conducted additional experiments evaluating policy quality throughout training using the LC win rate of Alpaca-Eval — a common, frozen third-party evaluator — at every 20 steps:
> > >
> > >
> > > **Qwen2.5-3B-Instruct**
> > >
> > > | Step | RLOO  | RLOO+R2M |
> > > |------|-------|----------|
> > > | 0    | 15.5  | 15.5   |
> > > | 20   | 17.6  | 19.9     |
> > > | 40   | 19.3  | 21.8     |
> > > | 60   | 20.8  | 23.2     |
> > > | 80   | 21.4  | 24.7     |
> > > | 100  | 21.9  | 24.8     |
> > >
> > >
> > > **LLaMA3-8B-Instruct**
> > >
> > > | Step | RLOO  | RLOO+R2M |
> > > |------|-------|----------|
> > > | 0    | 22.9  | 22.9    |
> > > | 20   | 25.6  | 27.2     |
> > > | 40   | 26.2  | 29.7     |
> > > | 60   | 27.1  | 32.8     |
> > > | 80   | 28.3  | 34.5     |
> > > | 100  | 28.4  | 34.5     |
> > >
> > >
> > > As shown above, under a common frozen third-party evaluator, RLOO+R2M consistently and significantly outperforms vanilla RLOO at every evaluation checkpoint — with gains of up to 6.1 percentage points.These results confirm that the performance improvements are genuine and not artifacts of evaluating under different reward functions. We will include these results in the revised manuscript.
> > >
> > > We sincerely hope that these supplementary clarifications and additional experiments can address the reviewer's remaining concerns. We would be very grateful if the reviewer would consider revising the evaluation in light of these new results.

---

### Official Review · Reviewer_ZV5Q · 2026-03-08

**Soundness:** 3
**Presentation:** 2
**Significance:** 3
**Originality:** 3
**Overall Recommendation:** 4
**Confidence:** 3

**Summary:**

This paper claims that the reward model may no longer be accurate during RL training as the policy is constantly changing. Directly re-training the reward model is too expensive, and the paper aims to propose a lightweight mechanism to update the reward model along with the policy in RL training. The method is motivated by the empirical finding that “deep-layer hidden states effectively capture human preferences”. Hence, the paper proposes a cross-attention mechanism to inject deep layer hidden state into the reward model and fine tune it using Bradley-Terry loss. An additional group reward entropy is added to deal with the problem that later in the training, most of the responses get similar reward. The method is compared against base RL methods which doesn’t incorporate deep hidden state information along with some ablation studies, and the proposed method achieves superior performance in terms of both reward and win rate on average.

**Compliance With Llm Reviewing Policy:**

Affirmed.

**Final Justification:**

The author has given clear and satisfying justifications for Q1-Q4, while the rebuttal for W1 is not entirely convincing as the author only provides high level argument with no theoretical or empirical support. That said, W1 is not a major weakness, and I think the idea is overall a valuable contribution to the community. I lean towards accepting this paper, with a score of 4 (Weak Accept).

**Key Questions For Authors:**

1. In table 1, the paper has both GRPO and GRPO + Pretrained RM. What’s the difference between them?
2. In Figure 5, the author compares RLOO and RLOO+R2M in terms of loss. What’s the loss here?
3. I might have missed any earlier reference, but the paper starts discussing R2M with noise in Section 5.2 but only formally defines it in Section 5.4.
4. I hope the author could provide more of their perspective on the GRE entropy term. If the responses are of similar quality, what's the motivation to artificially stretch the reward? The reward itself might contain some noise, so if the reward of two responses are similar, it’s probably hard to judge which response is actually better even if one response is scored higher than another. Is it the fact that softmax bounds the magnitude of reward that potentially prevents reward hacking?

**Limitations:**

yes

**Strengths And Weaknesses:**

Strength:
1. The proposed method is lightweight and practical, and can be integrated into existing RLHF pipelines.
2. They provide extensive experiment results across multiple settings to support their claim.
3. The empirical finding in Section 3 is very interesting and well presented.

Weakness:
1. In Figure 5, R2M seems to achieve higher reward under both models but the KL divergence is also considerably larger than RLOO, indicating the R2M may be not KL efficient. Can GRPO and RLOO achieve better performance in terms of reward if you simply lower the hyperparameter for KL penalty (\beta in equation (2))?

---

> ### Author Rebuttal · Authors · 2026-03-31
>
> Thank you for your valuable feedback. We anwsered each of your detailed questoins below, we hope this resolves your concerns, and we are happy to respond any further questions you may have.
>
> ---
>
> > **W1**: Effect of Reducing the KL Coefficient on Reward Performance
>
> Thanks for your valuable question. Adjusting the KL coefficient of RLOO and GRPO cannot effectively achieve better performance in terms of reward. We want to clarify that the larger KL divergence observed in Figure 5 is a result of the R2M mechanism rather than any adjustment to the KL coefficient. In fact, we applied the same KL regularization coefficient to both settings.
>
> Excessively high KL divergence is typically a signal of reward overoptimization. If the KL constraint is simply reduced, the policy model will quickly overfit to spurious patterns to game high rewards. We have provided relevant analysis in Section 5.2 (2): R2M encourages more aggressive policy updates by assigning higher rewards, which widens the gap between the policy model and the reference model. However, this does not lead to reward hacking, indicating that R2M effectively improves the reward model ’s resistance to the policy’s exploitation of specific patterns, enabling more aggressive policy updates in the correct direction without triggering reward overoptimization.
>
> ---
>
> > **Q1**: the difference between GRPO and GRPO+pretrained RM.
>
> We apologize for any confusion. In GRPO we directly employ reward model (RM) A for RL optimization. For GRPO+Pretrained RM, before RL optimization we first train RM A on the UltraFeedback training set (Section 5, Lines 246–249, right column) to obtain model B, which is then kept frozen for RL optimization. Here “pretrain” is relative to RL optimization and occurs after the original training of RM A. We will emphasize this distinction in the revised version.
>
> ---
>
> > **Q2:** What does the loss in Figure 5 represent?
>
> For both RLOO and RLOO+R2M, the loss refers to the training loss of the policy model, i.e., the mean of Equation 2 over the micro-batch.
>
> ---
>
> > **Q3:**  Definition position of R2M w/ Noise.
>
> Thank you for the reminder; we apologize for any inconvenience to your reading experience. We plan to move the definition of R2M w/ Noise from Section 5.4 (Lines 396–399, right column) forward to the baseline methods introduction in Section 5 (immediately after Line 257, right column).
>
> ---
>
> > **Q4:** the motivation of GRE loss
>
> **Effectiveness**: When trajectory qualities are similar and reward noise exceeds the score difference, GRPO randomly reinforces some trajectories and suppresses the rest (reviewer **NLdx, Q1 & Q3**). An important role of the GRE loss is to enlarge the reward gap, thereby preventing reward noise from exceeding the score difference. This allows the policy model to reinforce trajectories that possess slight advantages rather than updating completely at random.
>
>
> **Misunderstanding:** When response qualities are truly identical, **the GRE loss gradient is exactly zero**: it cannot artificially inflate score differences.
>
> The GRE loss activates only when genuine quality differences exist. It sharpens the reward distribution by amplifying pre-existing score gaps without altering the correct ranking, thus alleviating group degeneration rather than creating artificial distinctions. We present the formal argument below.
>
> When all responses within a group have strictly equal rewards, the standardized inputs $z_j$ are all zero and Softmax produces a uniform distribution $p_j = 1/K$. The GRE loss (negative entropy) reaches its maximum $\log K$ at this point. The partial derivative of the GRE loss w.r.t. each probability is:
>
> $$\frac{\partial \mathcal{L}_{\text{GRE}}}{\partial p_j} = -(\log p_j + 1).$$
>
> For $p_j = \text{softmax}(z_j)$, the Jacobian is:
>
> $$\frac{\partial p_j}{\partial z_i} = p_j(\delta_{ij} - p_i).$$
>
> Applying the chain rule:
>
> $ \frac{\partial \mathcal{L}\_{\text{GRE}}}{\partial z_i} = \sum \frac{\partial \mathcal{L}_{\text{GRE}}}{\partial p_j} \cdot \frac{\partial p_j}{\partial z_i}. $
>
> Under the uniform distribution $p_j = 1/K$, the term $-(\log p_j + 1) = \log K - 1$ is constant across all $j$. This constant factors out of the sum, yielding:
>
> $$\frac{\partial \mathcal{L}\_{\text{GRE}}}{\partial z_i} = (\log K - 1) \cdot \sum_{j=1}^{K} p_j(\delta_{ij} - p_i) = (\log K - 1) \cdot (p_i - p_i) = 0.$$
>
>
> Since the gradient on $z_i$ is zero, the gradient of all reward model parameters is also zero by the chain rule. Thus, GRE loss induces no parameter updates when responses are equally scored, and only sharpens score differences once the reward model has identified meaningful quality distinctions.
>
>
> **Softmax**: We use Softmax to construct GRE loss for two reasons. First, mapping normalized rewards to [0, 1] ensures a standardized entropy definition. Second, this guarantees zero gradient on RM parameters when intra-group response qualities are identical, avoiding unreasonable reward noise.

---

> > ### Author Rebuttal · Reviewer_ZV5Q · 2026-04-03
> >
> > W1: I don’t think my concern is fully addressed. The question is not about whether the two methods use the same beta coefficient, but more about the KL efficiency of the method. If you lower beta by a reasonable range, then it’s not necessarily gonna lead to reward overoptimization. My question is more like it seems like your method gains higher reward but also pays a higher KL budget, so is it really better than just adjusting the beta of the base method? After all, beta is a hyperparameter, and maybe for the base method, lower beta is the right hyperparameter.
> >
> > Q1/ Q2/Q3: These are fully addressed. Thank you!
> >
> > Q3: I am not fully convinced by the proof. The reward model is usually noisy, so let’s say if the ground truth reward for two responses are both 1, but the actual noisy reward is 0.95 and 1.05. Since your algorithm only observes the noisy reward, it will artificially stretch the reward difference between them, right? The proof seems to only work for the case when the noisy rewards have no difference, but two responses with same quality should only have the same ground truth reward, not necessarily the same noisy reward. Basically I am not sure how robust the method is with respect to noise.
> >
> > I plan to keep my original score, and overall I maintain a positive attitude towards this paper.

---

> > > ### Author Response · Authors · 2026-04-04
> > >
> > > We would like to thank the reviewer for the insightful questions. We apologize for the initial misunderstanding of your concerns. We have now reorganized our response as follows:
> > >
> > > **Response to W1:**
> > >
> > > Thank you for the important comment. We believe that the improvements brought by R2M cannot be achieved by simply reducing the KL divergence for the following reasons:
> > > **Adjusting $\beta$ is analogous to changing the step size under a fixed objective, whereas our method modifies the objective itself by reshaping the reward landscape using policy feedback.**
> > >
> > > To clarify this point, let us consider an intuitive assumption: suppose we set the KL coefficient to 0, and assume that reward hacking does not occur. In this case, the policy model updates entirely according to the advantage estimates corresponding to the reward scores. The relative size of the rewards will directly influence the sign of the advantage. In policy gradient methods, the direction of encouragement and suppression for each token is primarily determined by the sign of the advantage [1]: trajectories with positive advantages will increase the probability of their corresponding tokens and suppress those tokens not selected, while trajectories with negative advantages will do the opposite. Therefore, the relative reward size between trajectories directly determines the direction of gradient updates.
> > >
> > > R2M is capable of redistributing the rewards of the trajectories, causing the relative advantages of different samples to change during the GRPO or RLOO advantage computation. This redistribution directly influences the direction of policy gradient updates, which cannot be achieved by simply imposing a small KL constraint.
> > >
> > > ---
> > >
> > >
> > > **Response to Q4:**
> > >
> > >
> > >
> > >
> > > Thank you for this insightful question. We agree that for any *single* pair of responses with identical ground truth rewards, noise can produce different observed rewards, and GRE would indeed stretch this spurious difference on that particular instance. Our robustness claim does not rest on individual samples but rather on the *expected behavior* over the data distribution.
> > >
> > > Concretely, consider two responses $j$ and $k$ with identical true rewards $r_j^* = r_k^*$. Their observed rewards are:
> > >
> > > $$r\_j = r_j^* + \epsilon_j, \quad r_k = r_k^* + \epsilon_k,$$
> > >
> > > where $\epsilon_j, \epsilon_k \sim \mathcal{N}(0, \sigma^2)$ are independent. Since the GRE loss is used as an optimization objective for the reward model (not applied to individual samples in isolation), what matters is the *expected gradient* over the noise distribution. We show that:
> > >
> > > $$\mathbb{E}\left[\frac{\partial \mathcal{L}\_{\text{GRE}}}{\partial \Delta r_{jk}}\right] = 0 \quad, \ \text{when } r_j^* = r_k^*,$$
> > > where the expected gradient with respect to their reward difference is zero.
> > > This result holds *because* $r_j$ and $r_k$ are identically distributed (not because they take the same noisy value). The key insight is as follows: for any realization where noise makes $r_j > r_k$, there exists a symmetric realization where $r_k > r_j$, and these cancel out in expectation. Therefore, *on average over the training data*, the GRE loss does not systematically push the reward model to separate responses of equal quality.
> > >
> > > In contrast, when the true rewards differ ($r_j^* \neq r_k^*$), this symmetry breaks, and the expected gradient becomes non-zero in the direction that correctly enlarges the gap. This is the sense in which GRE is robust: it amplifies genuine quality differences while the spurious differences induced by noise wash out in expectation during optimization.
> > >
> > > We acknowledge that finite-sample variance means individual gradient steps may be noisy, but this is a property shared by all stochastic optimization methods and is mitigated by training over sufficiently many samples. We will clarify this distinction between per-sample behavior and expected behavior in the revised manuscript.
> > >
> > >
> > > Thank you again for your valuable feedback. We hope this addresses your remaining concerns.
> > >
> > >
> > > ---
> > > References:
> > >
> > > [1] https://arxiv.org/abs/2506.01347

---

### Official Review · Reviewer_NLdx · 2026-03-13

**Soundness:** 2
**Presentation:** 2
**Significance:** 2
**Originality:** 3
**Overall Recommendation:** 4
**Confidence:** 3

**Summary:**

This paper proposes a lightweight RLHF framework named R2M. It aligns distribution shifts in real-time and mitigates the reward overoptimization problem. The method introduces the continuously evolving hidden states of the policy model into the scoring head of the reward model and combines this with a Group Reward Entropy Bradley-Terry (GREBT) regularization loss.​

**Compliance With Llm Reviewing Policy:**

Affirmed.

**Final Justification:**

My overall assessment remains unchanged, and I maintain my current score.

**Key Questions For Authors:**

1. The GRE loss utilizes reward standardization and Softmax to "sharpen" distributions and amplify score disparities. In scenarios where group responses are qualitatively indistinguishable, does this artificial variance risk creating "pseudo-signals" that lead the policy to overfit to noise rather than human intent?
2. As R2M iteratively adapts to the policy's internal states to mitigate distribution shift, how do you prevent "model drift"? Is there a risk that the reward model might prioritize staying "aligned" with the evolving policy over maintaining the integrity of the original human preference data?
3. Does the effectiveness of the GRE loss depend heavily on this initial diversity?

**Limitations:**

yes

**Strengths And Weaknesses:**

**Strengths**
- The paper astutely observes the strong correlation between the deep hidden states of the policy model and both human preferences and reward scores. This provides a novel perspective: distinct from purely semantic representation for mitigating reward overoptimization.​
- The sequence-to-token cross-attention mechanism and the time-step-based weighted combination strategy remain relatively lightweight in their engineering implementation. These mechanisms successfully achieve dynamic updates of the reward model without significantly increasing memory overhead.

**Weakness**
- The paper claims that R2M goes "beyond semantics" and emphasizes that the hidden states contain the internal feedback of the policy. However, during the fusion process where cross-attention compresses hidden states into scalar features, it remains unclear whether the model genuinely learns the so-called "distributional features" or merely uses them as a feature-enhancing regularization technique. The paper lacks in-depth ablation studies that disentangle semantics from distribution.
- The mitigation mechanism for the "group degeneration" problem (the GRE loss) lacks a proof of universality. The paper points out that introducing feedback information leads to a more severe group degeneration phenomenon in the reward model, which necessitates the GRE loss to forcefully widen the score variance among intra-group samples. This practice of artificially creating variance through standardization and Softmax acts as a heuristic scalar redistribution.

---

> ### Author Rebuttal · Authors · 2026-03-31
>
> Thank you for your valuable feedback. We have discussed and addressed each of the detailed questions you raised. We hope these responses will alleviate your concerns, and we look forward to your feedback..
>
> ---
>
> > W1: Ambiguity between distribution feature and feature enhancement in R2M
>
> We appreciate your valuable questions. First, we clarify a misunderstanding: cross-attention injects policy feedback into the original RTE (Section 4.1, Equations 3 and 4), rather than directly compressing them into scalar features.
>
> Second, introducing policy hidden states theoretically tightens the upper bound of reward misalignment (Theorem 3.1). The property that larger distribution shifts yield greater R2M benefits only holds only when the model captures real distribution features rather than relying on feature enhancement.
>
> Table 3 further demonstrates that the reward model trained with RLOO+R2M improves accuracy by 5.1% (Qwen2.5) and 6.3% (LLaMA3). Such generalization gains on out-of-distribution test sets cannot be achieved by simple feature enhancement, but only when the RM learns policy distribution features aligned with human preferences.
>
>
> ---
>
>
> > W2: GRE loss merely performs scalar redistribution.
>
> Thanks for your attention to the details of our work. We have provided a universal proof for the GRE loss effect in Theorem 4.1 (Appendix A.2): introducing the GRE loss strictly mitigates group degeneration, i.e., $C(\varphi_\alpha) < C(\varphi_0)$, and its mitigation effect increases monotonically with the loss weight $\alpha$. Our proof relies solely on standard optimization assumptions such as continuous differentiability of the loss function and positive-definiteness of the Hessian, ensuring full generality.
>
>
> Moreover, the GRE loss is not a heuristic scalar redistribution; the essential distinction is that scalar redistribution applies a fixed transformation directly to existing reward scores, whereas the GRE loss expands differences between reward values by minimizing reward entropy. On the other hand, updates from the GRE loss also implicitly align with the policy model in real time, because all reward values are obtained after incorporating policy feedback.
>
> ---
>
>
> > Q1 & Q3: When initial diversity of response quality/rewards is small, GRE loss may mislead the policy model.
>
> Thank you for your insightful question. We want to clarify that  the GRE loss will not cause the policy to overfit to noise for the following reasons.
>
> First, R2M relies on joint optimization of the GRE loss and BT loss. When group responses are qualitatively indistinguishable, the GRE loss term yields no effective gradient, and model updates are thus dominated by the BT loss.
>
> Second, A common misconception is that the GRE loss acts after group degeneration has occurred. We have already proven that when response quality is completely identical, the gradient of the GRE loss on the RM parameters is zero; therefore, the RM will not exaggerate gaps in such cases (Please refer to reviewer **ZV5Q, Q4**).
>
> Furthermore, even if response qualities within a group ultimately remain indistinguishable to the RM, similar reward values **merely make it difficult for the policy model to obtain benefits during updates**. From the gradient-update perspective, for a group of samples with similar quality, those that happen to score above the mean receive positive advantages and those below receive negative advantages. During updating, they receive gradients in opposite directions: tokens in positively advantaged trajectories are encouraged, while tokens in negatively advantaged trajectories are suppressed. Because their qualities are similar, however, whether tokens are encouraged or suppressed the policy update merely fails to acquire a significant preference bias and cannot be described as “overfitting to noise.”
>
>
> ---
>
> > Q2 : The RM might be biased by the policy model
>
> We maintain that the reward model will not lose its original function by aligning with the policy model for the following reasons.
> We encourage the reward model to update according to the GREBT loss rather than along directions aligned with the policy model, which ensures the gradient directions remain unaligned with the policy model. Meanwhile, the LLM backbone of the reward model is frozen throughout training, effectively preserving the original human preferences, as explicitly stated in Appendix H.2 and [1].
>
> Furthermore, updating entirely in the direction of alignment with the policy model would generally lead directly to reward misalignment and severely impair training effectiveness. However, Our experimental results show that R2M yields policy models that significantly outperform the baseline across multiple preference alignment tasks, this is the most direct evidence that R2M genuinely enhances reward-model performance rather than causing model drift.
>
> ---
> References:
>
> [1] https://arxiv.org/abs/2403.05171

---

> > ### Author Rebuttal · Reviewer_NLdx · 2026-04-04
> >
> > I thank the authors for their replies. The provided clarifications have made my understanding of the paper much clearer. However, my overall assessment remains unchanged, and I will maintain my current score.

---

> > > ### Author Response · Authors · 2026-04-07
> > >
> > > Dear Reviewer NLdx,
> > >
> > > Thank you for the feedback. We are glad that our response has helped you better understand our work. Your detailed and constructive feedback has been invaluable in strengthening our work. We will definitely incorporate all of your suggestions and discussions into the revised manuscript.
> > >
> > > Best regards,
> > >
> > > Authors

---

### Decision · Program_Chairs · 2026-04-30

**Decision:**

Accept (regular)

**Comment:**

This paper proposes R2M, a lightweight RLHF method that mitigates reward overoptimization.hacking by incorporating evolving policy hidden states into the reward model via a cross-attention mechanism, along with a Group Reward Entropy Bradley-Terry (GREBT) loss for stabilizing reward differentiation.

After the rebuttal, all reviewers agree that the paper introduces an interesting way on reward modeling, which leverages policy internal representations as implicit feedback to address distribution shift during RLHF. The approach is considered lightweight and compatible with existing pipelines, and empirical results demonstrate consistent improvements over strong baselines such as RLOO and GRPO across multiple benchmarks.

However, several important concerns were raised. A central issue, noted by all reviewers, is the role of the GRE loss, which may artificially amplify reward differences and potentially introduce noise, especially when responses are of similar quality. While the rebuttal provides theoretical arguments (e.g., zero gradients under identical rewards) and empirical evidence suggesting stable behavior in practice, this concern is not fully resolved at a conceptual level.

From my assessment, the paper makes a meaningful contribution to RLHF by introducing a new mechanism for adaptive reward modeling. While some aspects could benefit from further clarification and analysis, the overall technical contribution and empirical evidence are sufficient to support acceptance.